# Lattice pinning in $MoO_3$ via coherent interface with stabilized $Li^+$ intercalation

Shuo Sun[1], Zhen Han[2], Wei Liu[1], Qiuying Xia[1], Liang Xue[1], Xincheng Lei[2], Teng Zhai [1] ✉, Dong Su [2] ✉ & Hui Xia [1] ✉

Large lattice expansion/contraction with $Li^+$ intercalation/deintercalation of electrode active materials results in severe structural degradation to electrodes and can negatively impact the cycle life of solid-state lithium-based batteries. In case of the layered orthorhombic $MoO_3$ ($\alpha$-$MoO_3$), its large lattice variation along the $b$ axis during $Li^+$ insertion/extraction induces irreversible phase transition and structural degradation, leading to undesirable cycle life. Herein, we propose a lattice pinning strategy to construct a coherent interface between $\alpha$-$MoO_3$ and $\eta$-$Mo_4O_{11}$ with epitaxial intergrowth structure. Owing to the minimal lattice change of $\eta$-$Mo_4O_{11}$ during $Li^+$ insertion/extraction, $\eta$-$Mo_4O_{11}$ domains serve as pin centers that can effectively suppress the lattice expansion of $\alpha$-$MoO_3$, evidenced by the noticeably decreased lattice expansion from about 16% to 2% along the $b$ direction. The designed $\alpha$-$MoO_3$/$\eta$-$Mo_4O_{11}$ intergrown heterostructure enables robust structural stability during cycling (about 81% capacity retention after 3000 cycles at a specific current of $2\,A\,g^{-1}$ and $298 \pm 2\,K$) by harnessing the merits of epitaxial stabilization and the pinning effect. Finally, benefiting from the stable positive electrode–solid electrolyte interface, a highly durable and flexible all-solid-state thin-film lithium microbattery is further demonstrated. This work advances the fundamental understanding of the unstable structure evolution for $\alpha$-$MoO_3$, and may offer a rational strategy to develop highly stable electrode materials for advanced batteries.

Owing to the fast development of electric vehicles and grid energy storage, there is increasing demand for lithium-ion batteries with long cycle life[1–3]. As compared to the negative electrode material of graphite, the positive electrode materials possess relatively lower specific capacity and shorter cycle life, which primarily limit the electrochemical performance of lithium-ion batteries[4–6]. To achieve high energy density and long cycle life at the same time, the positive electrode materials need to be able to accommodate a large amount of lithium ions without imposing permanent change to the crystal structure. Large lattice variations accompanied with $Li^+$ intercalation/deintercalation will degrade the structure of positive electrode materials, which could induce irreversible phase transition and shorten cycle life. Especially for the solid-state lithium batteries, large volume variation in the positive electrode materials could lead to a breach of the electrode–solid electrolyte interface, resulting in large interfacial impedance and battery failure[7,8]. However, there is always a trade-off between large specific capacity and cycle life for positive electrode materials, and developing stabilized intercalation positive electrodes that can accommodate a large quantity of lithium ions is still a challenge.

Among the various positive electrode materials, layered $\alpha$-$MoO_3$ is particularly appealing to lithium ion insertion due to its edge-sharing $MoO_6$ octahedra bilayers stacked along [010] by van der Waals (vdW)

[1]School of Materials Science and Engineering, Nanjing University of Science and Technology, 210094 Nanjing, PR China. [2]Beijing National Laboratory for Condensed Matter Physics, Institute of Physics, Chinese Academy of Sciences, Beijing, China. ✉e-mail: tengzhai@njust.edu.cn; dongsu@iphy.ac.cn; xiahui@njust.edu.cn

interaction[9]. A high theoretical specific capacity of about 372 mAh g$^{-1}$ is expected, corresponding to the Mo$^{6+}$/Mo$^{4+}$ redox couple with two lithium storage per Mo, which makes α-MoO$_3$ a promising positive electrode for Li-ion storage[10–12]. However, the insertion of Li$^+$ ions into the layered α-MoO$_3$ breaks the thermodynamically stable state and induces large lattice expansion (~16%), resulting in the irreversible phase transformation to acicular Li$_x$MoO$_3$ (x ~0.25) and fast capacity fading (Fig. 1a)[13,14]. To prolong the cycle life of α-MoO$_3$, various strategies including constructing nanostructures[15], nitrogen doping[16], and widening the vdW interlayer[10,13] of α-MoO$_3$ have been extensively explored. Nevertheless, there is still notable capacity loss, especially during the initial cycles for the modified α-MoO$_3$[10,13], suggesting more effective approaches need to be developed to further stabilize the layered structure without compromising its large specific capacity.

Recent studies on constructing spinel-layered heterostructures for LiMnO$_2$ and Li-rich (Li$_2$MnO$_3$) electrode materials suggest that interface engineering could be an effective way to improve the structural stability of Li$^+$ intercalation host[17,18]. To reduce the large lattice expansion of α-MoO$_3$ during the lithiation process, we propose a coherent epitaxial interface pinning design, which can inhibit the mobility of Mo atoms via strong ionic bonds at coherent interfaces. In this study, we constructed α-MoO$_3$/η-Mo$_4$O$_{11}$ intergrown heterostructure with coherent epitaxial

interface by magnetron sputtering. η-Mo$_4$O$_{11}$ is a nearly "zero-strain" Li$^+$ intercalation host (Fig. 1b), which could be the perfect structural stabilizer for the coherent interface design (Fig. 1c). When a coherent interface can be established between α-MoO$_3$ and η-Mo$_4$O$_{11}$, the high structural stability of η-Mo$_4$O$_{11}$ can be transferred to α-MoO$_3$, which could suppress lattice variation of α-MoO$_3$ during lithiation and delithiation. Therefore, the coherent interface pinning design could provide an effective structural regulation strategy to stabilize electrode materials with intercalation chemistry.

Herein, pure α-MoO$_3$, α-MoO$_3$/η-Mo$_4$O$_{11}$, and pure η-Mo$_4$O$_{11}$ thin films were prepared respectively by manipulating the O$_2$/Ar flow ratio during the sputtering process. With the coherent interface, the large lattice expansion and phase transition in α-MoO$_3$ have been viably suppressed during the lithiation, which was evidenced by the ex situ XRD. The density functional theory (DFT) calculations indicate the interface pinning can effectively improve the structural stability of α-MoO$_3$ and suppress Mo migration during charge/discharge. Consequently, the α-MoO$_3$/η-Mo$_4$O$_{11}$ heterostructured electrode achieved a reversible specific capacity as high as ~350 mAh g$^{-1}$ (0.1 A g$^{-1}$; 1.0–3.5 V vs. Li/Li$^+$) and a high capacity retention of about 81% after 3000 cycles (at a specific current of 2 A g$^{-1}$ and 298 ± 2 K) in the organic electrolyte, demonstrating greatly enhanced structural stability of α-MoO$_3$ within

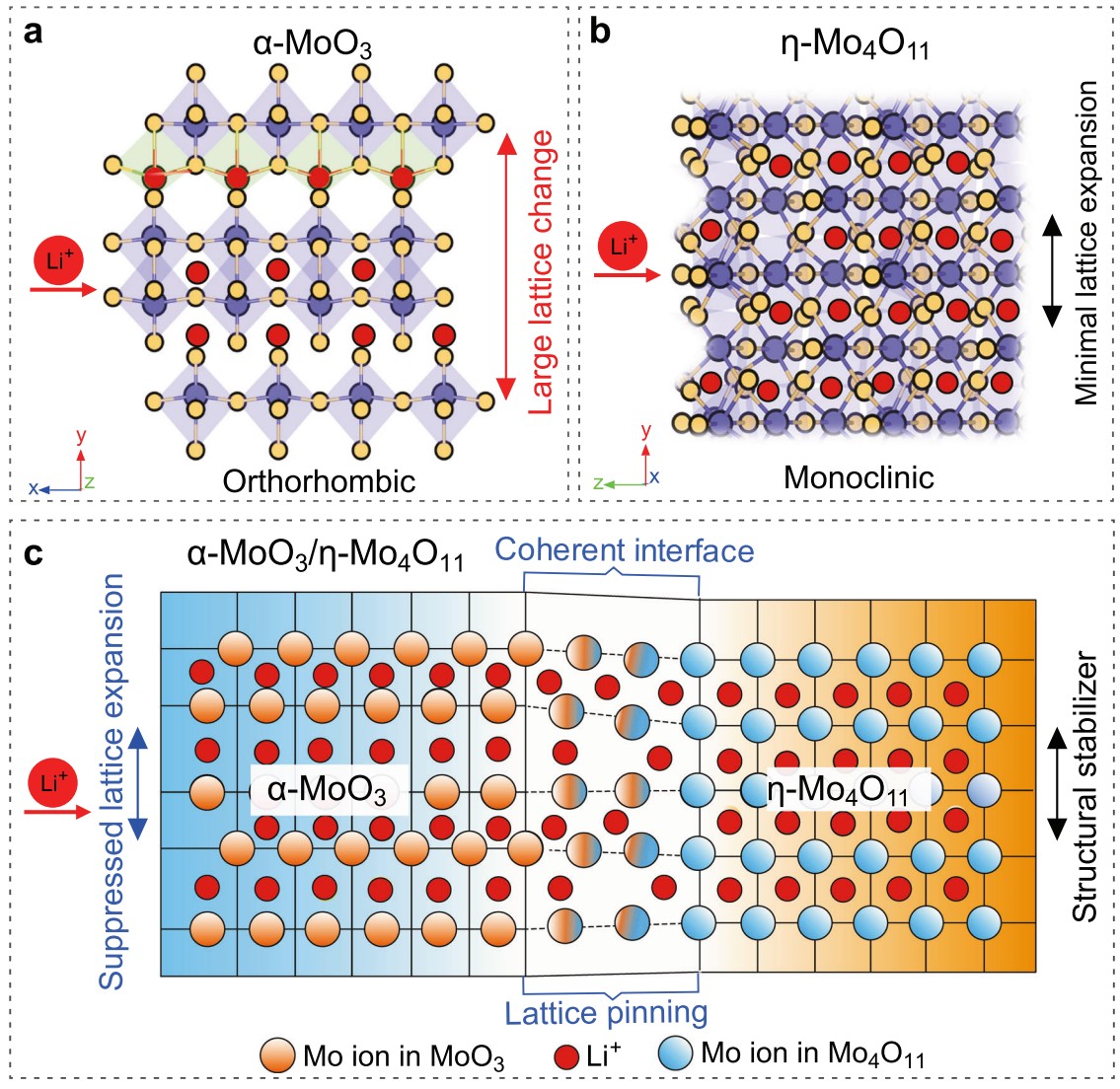

**Fig. 1 | Coherent interface design to suppress lattice change of layered MoO$_3$ during lithiation.** Schematic illustrations of **a** large lattice expansion of the individual α-MoO$_3$, **b** minimal lattice expansion of the individual η-Mo$_4$O$_{11}$,

and **c** suppressed lattice expansion of α-MoO$_3$ via coherent interface in the α-MoO$_3$/η-Mo$_4$O$_{11}$ heterostructure during lithiation.

the heterostructure. Finally, we successfully fabricated an all-solid-state thin-film lithium microbattery based on the α-MoO₃/η-Mo₄O₁₁ electrode, lithium phosphorous oxynitride (LIPON) solid electrolyte, and Li metal electrode. Benefiting from the stable positive electrode−solid electrolyte interface during Li⁺ intercalation/deintercalation processes, the microbattery showed a long-term cycling stability with about 74% capacity retention after 4000 cycles (at a specific current of 2 A g⁻¹ and 298 ± 2 K).

## Results

### Construction of α-MoO₃/η-Mo₄O₁₁ heterostructure and structural characterizations

The fabrication of α-MoO₃, α-MoO₃/η-Mo₄O₁₁, and η-Mo₄O₁₁ thin films on Pt/Ti/glass and Pt/Ti/stainless steel substrates through direct current (DC) magnetron sputtering under O₂/Ar atmosphere is schematically illustrated in Fig. 2a. During the sputtering process, the as-formed ionized Ar (Ar⁺) are driven towards the Mo target under the electric field. After Ar⁺ bombardment, the target Mo atoms are released to react with the O⁻ at ~400−800 K and deposit on the substrate in the form of Mo oxides[19]. X-ray diffraction (XRD) and corresponding Rietveld refinement were carried out to identify phase evolution and crystal structure of the films under different deposition conditions. The O₂/Ar flow ratios of 8%, 13%, and 25% at a deposition temperature of 573 K were selected to prepare α-MoO₃, α-MoO₃/η-Mo₄O₁₁, and η-Mo₄O₁₁ thin films, respectively, which are demonstrated by the powder XRD and glancing incidence XRD (GIXRD) patterns (Supplementary Figs. 1 and 2). Specifically, via Rietveld refinement, Fig. 2b reveals that the heterostructure possesses approximately 63 wt% α-MoO₃ and 37 wt% η-Mo₄O₁₁, respectively, indicating α-MoO₃ is the major phase for the heterostructure. The refined phase fraction error for η-Mo₄O₁₁ in the heterostructure is 1.3% as provided by the MAUD. The Rietveld-refined XRD patterns demonstrate a monoclinic structure for η-Mo₄O₁₁ (Supplementary Fig. 3, JCPDS No. 13−0142) and an orthorhombic structure for α-MoO₃ phase (Fig. 2c and Supplementary Table 1, JCPDS No. 05−0508)[13,20]. Moreover, the chemical composition and Mo valence state of the film can be well controlled by tuning the oxygen partial pressure and substrate temperature during deposition. As shown in Fig. 2d, when the O₂/Ar flow ratio was larger than 20%, Mo atoms can be completely oxidized to form α-MoO₃. With the decrease of O₂/Ar flow ratio to less than 20%, low valence molybdenum oxides such as η-Mo₄O₁₁ were gradually generated together with α-MoO₃, resulting in the formation of α-MoO₃/η-Mo₄O₁₁ heterojunction. Such heterojunction design can be easily achieved by controlling the O₂/Ar flow ratio (10−20%) and substrate temperature (573−673 K). The standard deviation obtained during refinement for α-MoO₃, α-MoO₃/η-Mo₄O₁₁, and η-Mo₄O₁₁ is shown in Supplementary Table 2. The value in

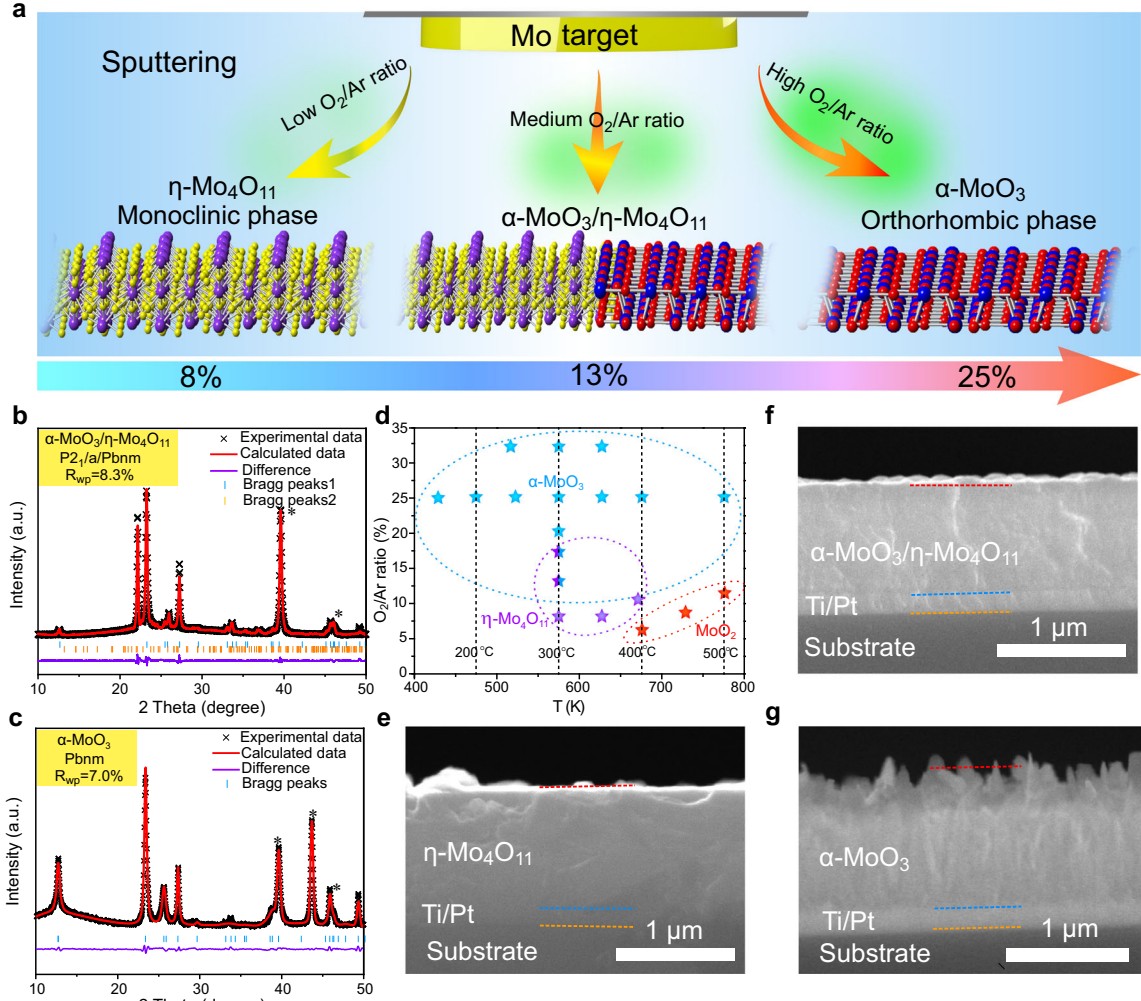

**Fig. 2 | Fabrication of the α-MoO₃/η-Mo₄O₁₁ heterostructure. a** Schematic diagram illustrating the fabrication of η-Mo₄O₁₁, α-MoO₃/η-Mo₄O₁₁, and α-MoO₃ thin films by magnetron sputtering. **b**, **c** Rietveld-refined XRD patterns of the α-MoO₃/η-Mo₄O₁₁ and α-MoO₃ thin films. The asterisk represents the substrate. **d** The corresponding O₂/Ar flow ratio versus the deposition temperature phase diagram. **e**–**g** Cross-section FESEM images of η-Mo₄O₁₁, α-MoO₃/η-Mo₄O₁₁, and α-MoO₃ thin films, respectively.

the bracket shows the standard deviation[21]. The phase evolution in thin films deposited under different $O_2$/Ar flow ratios is further confirmed by Raman measurements (Supplementary Fig. 4). The Raman spectrum of the $MoO_3$/η-$Mo_4O_{11}$ film was found to contain all the characteristic Raman bands of both α-$MoO_3$ and η-$Mo_4O_{11}$, demonstrating the successful fabrication of the heterostructure in the film by magnetron sputtering[22,23].

The surface morphologies of various samples were investigated by field-emission scanning electron microscopy (FESEM). As shown in Supplementary Fig. 5a–c, the increment of oxygen partial pressure not only induces the phase transition from η-$Mo_4O_{11}$ to α-$MoO_3$ but also increases the surface roughness of the film. At a low $O_2$/Ar flow ratio, the film exhibits a highly dense and smooth surface morphology, whereas at a high $O_2$/Ar flow ratio, the film is composed of nanoflakes, resulting in a rough surface morphology. Therefore, increasing the $O_2$/Ar flow ratio can enlarge particle size, change phase composition, and enhance the valance state of Mo ions of the films. The cross-section FESEM images of α-$MoO_3$, α-$MoO_3$/η-$Mo_4O_{11}$, and η-$Mo_4O_{11}$ thin films are presented in Fig. 2e–g, which reveal that the corresponding thicknesses of different films are about 1.1, 1.0, and 1.3 μm, respectively.

To investigate the heterostructure and interface between α-$MoO_3$ and η-$Mo_4O_{11}$, high-resolution transmission electron microscopy (HRTEM) and advanced spherical aberration-corrected scanning transmission electron microscopy (STEM) were employed for structural analysis. Ideal structure models of α-$MoO_3$ and η-$Mo_4O_{11}$ superimposed on the high-angle annular dark field (HAADF)-STEM images were plotted to better illustrate the intergrowth of α-$MoO_3$ and η-$Mo_4O_{11}$ phases and the formation of heterojunction. The atomic-resolution HAADF-STEM images of α-$MoO_3$ (Fig. 3a) and η-$Mo_4O_{11}$ (Fig. 3b) samples reveal lattice spacing of ~0.69 nm and 0.28 nm, respectively, which correspond to (020) plane of layered α-$MoO_3$ and

(020) plane of η-$Mo_4O_{11}$. Noted that the distances between Mo atoms along [010] direction are not all the same due to the bilayer structure of α-$MoO_3$. The corresponding lattice spacing of (020) plane of α-$MoO_3$ thus can be classified as the intralayer and interlayer spacing. The structures shown in Fig. 3a, b agree well with Mo atomic arrangements of the orthorhombic $MoO_3$ along [101] zone axis and monoclinic $Mo_4O_{11}$ along [100] zone axis (Fig. 3c). Supplementary Fig. 6 shows a low-magnification HAADF-STEM image of the α-$MoO_3$/η-$Mo_4O_{11}$ sample, revealing that the thin film is composed of nanograins with irregular shapes. An interfacial region in the α-$MoO_3$/η-$Mo_4O_{11}$ heterostructure is shown in Fig. 3d, in which η-$Mo_4O_{11}$ nano-domains with diameters of ~3–5 nm are found to be embedded in the layered α-$MoO_3$ matrix (Fig. 3d and Supplementary Fig. 7). The atomic arrangement of Mo within the blue square agrees well with the crystal model of monoclinic η-$Mo_4O_{11}$ along the [100] zone axis, whereas the atomic arrangement of Mo within the red square conforms to the crystal model of the layered α-$MoO_3$ along the [101] zone axis (Fig. 3d). Typically, the α-$MoO_3$/η-$Mo_4O_{11}$ heterostructure demonstrates a specific orientation relationship of $(020)_α$//$(020)_η$ and $[10-1]_α$//$[001]_η$ with a coherent interface between α-$MoO_3$ and η-$Mo_4O_{11}$ (Fig. 3d). Generally, the direct joining between α-$MoO_3$ (020) and η-$Mo_4O_{11}$ (020) planes (or a sharp interface) will generate a large lattice mismatch ($f$) of ~18.8%, defined by $f = (d_α - 2d_η)/d_α$, which is energetically unfavorable to form a coherent interface[24]. However, the HAADF-STEM image at the interfacial region demonstrates that a gradual transition zone with a thickness of ~4–5 nm instead of a sharp interface separates the two phases, thereby enabling the gradual release of misfit strain. In contrast to the sharp interface, the gradual transition interface enables the gradual release of misfit strain between the two lattices, leading to a continuous lattice mismatch less than 5% at the interfacial region. Therefore, the gradual transition interface is critical to reducing lattice

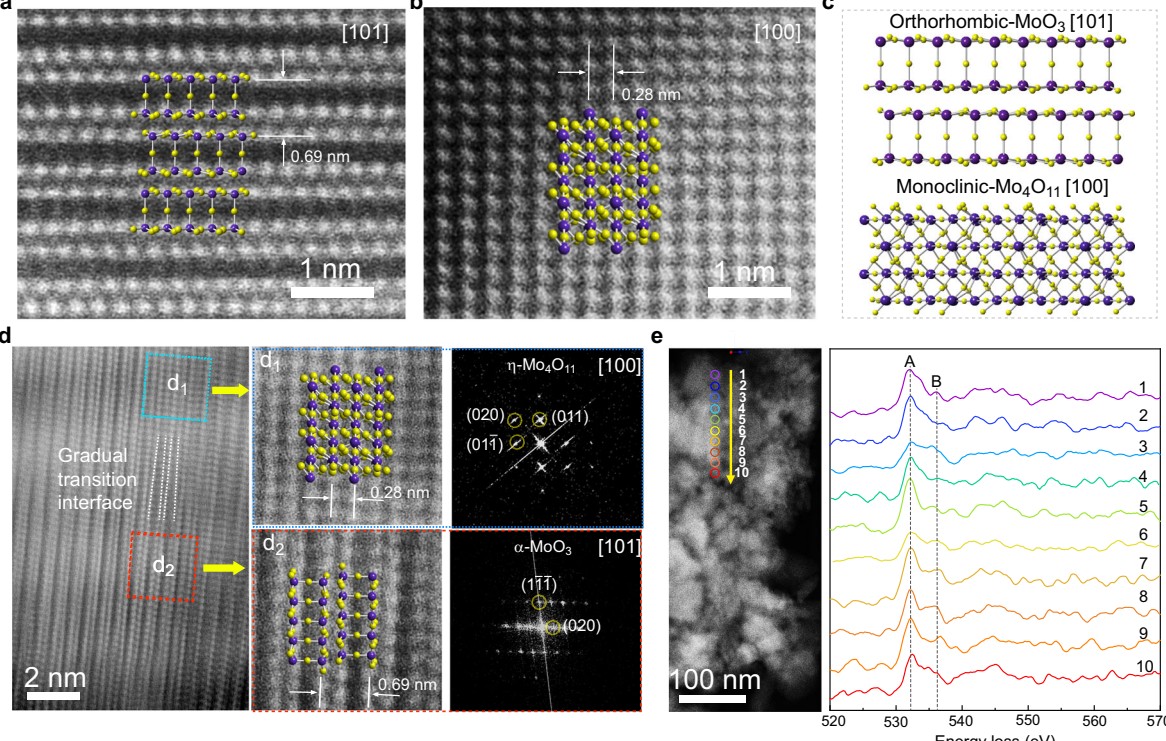

**Fig. 3 | Interface for the α-$MoO_3$/η-$Mo_4O_{11}$ heterostructure. a** HAADF-STEM image for layered α-$MoO_3$ along [101] zone axis. The purple spheres represent Mo atoms and the yellow spheres represent oxygen atoms. **b** HAADF-STEM image for monoclinic η-$Mo_4O_{11}$ along [100] zone axis. **c** Crystal structures of orthorhombic $MoO_3$ along [101] zone axis and monoclinic $Mo_4O_{11}$ along [100] zone axis. **d** HAADF-STEM image for the α-$MoO_3$/η-$Mo_4O_{11}$ heterostructure. The blue square corresponds to the monoclinic $Mo_4O_{11}$ viewed along the [100] zone axis while the red square corresponds to the orthorhombic $MoO_3$ viewed along the [101] zone axis. **e** The O $K$-edge ELNES spectra of the α-$MoO_3$/η-$Mo_4O_{11}$ heterostructure obtained along the yellow arrow in the left STEM image.

mismatch between α-MoO₃ and η-Mo₄O₁₁, resulting in the construction of coherent interface between these two phases. Simultaneously, the large stress and strain issues associated with sharp interface can be effectively mitigated by the gradual transition interface design, further improving the structural stability of the heterostructure.

Electron energy-loss spectroscopy (EELS) as a sensitive tool was employed to probe the oxidation state of Mo in the heterostructure. As shown in Fig. 3e, along the yellow line in the HAADF-STEM image, a series of energy-loss near-edge structures (ELNES) spectra of the O *K*-edge were recorded. The ELNES of O *K*-edge represents the transition of oxygen *s* core states to oxygen *p* states hybridized with Mo *4d* orbitals[25-27]. In Mo−O octahedral configuration, the crystal field splitting leads to the generation of $t_{2g}$ and $e_g$ symmetry bands, corresponding to peak A at about 532 eV and feature B at about 536 eV, respectively. For α-MoO₃, both the $t_{2g}$ and $e_g$ orbitals are empty, and thus O 1s electrons can be excited into both orbitals. Consequently, the corresponding two peaks can be clearly detected at the O *K*-edge ELNES spectrum of MoO₃. When the $t_{2g}$ orbitals in η-Mo₄O₁₁ are filled with electrons, an intensity decrease in peak A can be observed in O K-edge spectrum[25]. From the series of O K-edge ELNES spectra on the marked sites of α-MoO₃/η-Mo₄O₁₁ in Fig. 3e, it can be observed that the spectra of 3 and 6 show obviously decreased intensity at peak A, suggesting embedded nano-domains of η-Mo₄O₁₁ in α-MoO₃ matrix. The evolution of Mo chemical state in different samples was also investigated by X-ray photoelectron spectroscopy (XPS), as shown in Supplementary Fig. 8. Based on the Mo *3d* XPS spectra, the α-MoO₃

sample mainly contains Mo⁶⁺, and the trace of Mo⁵⁺ suggests the existence of oxygen vacancy. In comparison, the Mo⁵⁺/Mo⁶⁺ ratio notably increases in the α-MoO₃/η-Mo₄O₁₁ sample as half Mo are 5+ in η-Mo₄O₁₁, further confirming the formation of heterostructure.

## Electrochemical performance

To compare the electrochemical behaviors, the α-MoO₃, α-MoO₃/η-Mo₄O₁₁, and η-Mo₄O₁₁ thin films were investigated in Li metal coin cells with excess Li and an electrolyte of lithium perchlorate (LiClO₄) in a mixture of ethylene carbonate and dimethyl carbonate. All the cells were charged and discharged between 1.0 and 3.5 V (vs. Li/Li⁺) at a specific current of 0.1 A g⁻¹. As shown in Fig. 4a, the initial charge/discharge curves of the Li//α-MoO₃ half-cell show one pair of major potential plateaus at about 2.4 V (vs. Li/Li⁺) with a large reversible capacity of ~300 mAh g⁻¹. Notably, the first discharge curve of the α-MoO₃ electrode displays a minor potential plateau at ~2.8 V vs. Li/Li⁺, corresponding to the small anodic peak in cyclic voltammetry (CV) curve in Supplementary Fig. 9, which can be attributed to the irreversible phase transition to the new phase of LiₓMoO₃ with Li⁺ entering the intralayer[10,13,14,28]. With such irreversible phase transition, the α-MoO₃ electrode exhibits fast capacity decay even within five cycles. As shown in Fig. 4b, the charge/discharge curves of the Li//η-Mo₄O₁₁ half-cell present two pairs of potential plateaus between 1.5 and 3.5 V vs. Li/Li⁺ (corresponding to the two pairs of redox peaks in CV curves in Supplementary Fig. 9), delivering a large reversible capacity of about 350 mAh g⁻¹. With α-MoO₃ as the major phase in the heterostructure,

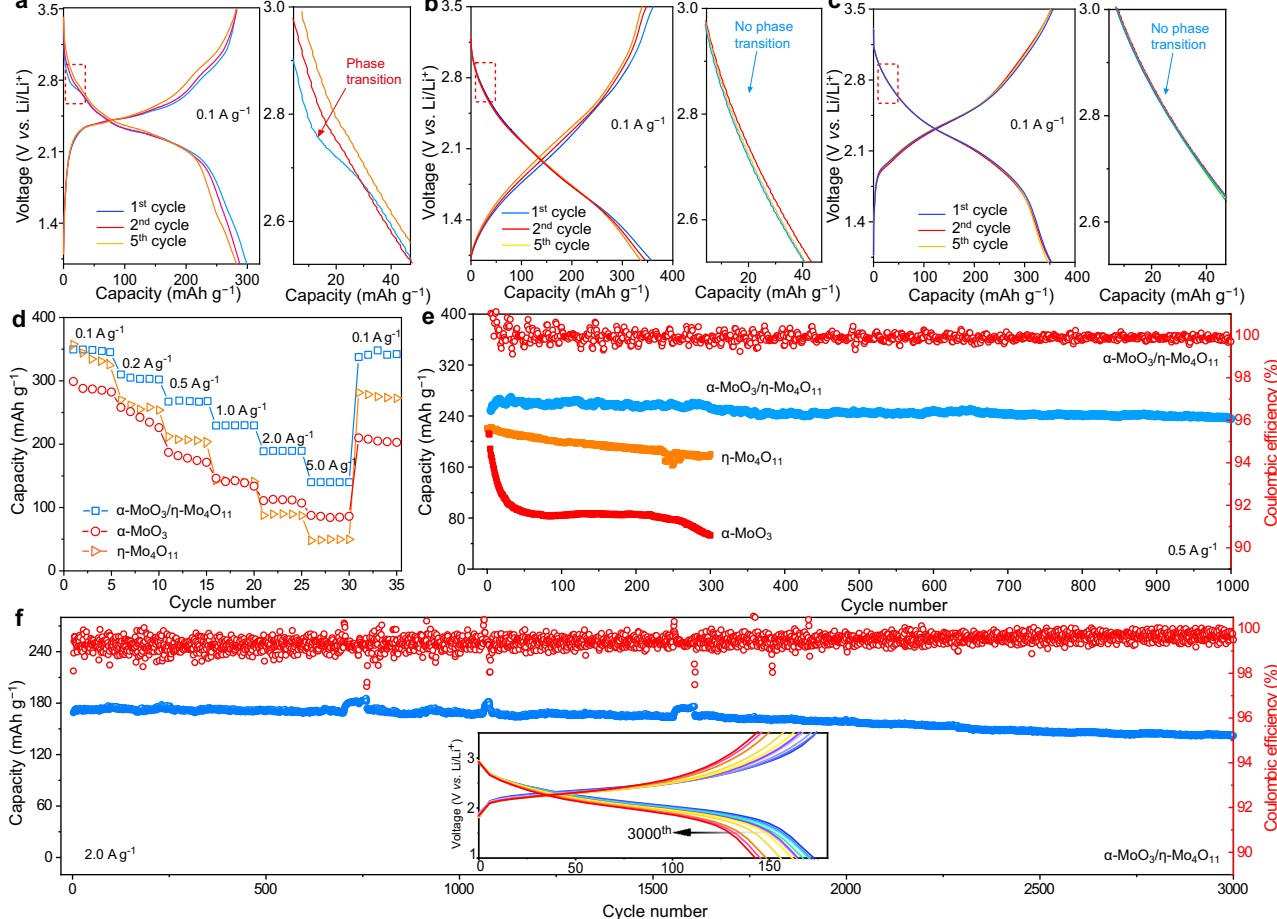

**Fig. 4 | Electrochemical behavior of the α-MoO₃/η-Mo₄O₁₁ heterostructure at 298 ± 2 K. a–c** Galvanostatic charge/discharge profiles of the α-MoO₃, η-Mo₄O₁₁, and α-MoO₃/η-Mo₄O₁₁ electrodes at the 1st, 2nd, and 5th cycles, respectively. **d** Rate capabilities of the α-MoO₃, α-MoO₃/η-Mo₄O₁₁, and η-Mo₄O₁₁ electrodes. **e** Cycle performance comparison of the α-MoO₃, α-MoO₃/η-Mo₄O₁₁, and η-Mo₄O₁₁ electrodes at 0.5 A g⁻¹. **f** Long-term cycle performance of the α-MoO₃/η-Mo₄O₁₁ electrode at 2.0 A g⁻¹. Inset shows the charge–discharge curves at different cycle numbers.

the charge/discharge curves of the $\alpha$-MoO$_3$/$\eta$-Mo$_4$O$_{11}$ electrode display similar potential plateaus as the $\alpha$-MoO$_3$ electrode but with a more sloping profile (Fig. 4c). A high reversible capacity of ~350 mAh g$^{-1}$ was achieved and well maintained by the $\alpha$-MoO$_3$/$\eta$-Mo$_4$O$_{11}$ electrode for the initial cycles. Surprisingly, the first discharge curve of the $\alpha$-MoO$_3$/$\eta$-Mo$_4$O$_{11}$ electrode maintains a steep slope at ~2.8 V vs. Li/Li$^+$ without the appearance of potential plateau, suggesting successful suppression of the irreversible phase transition in the layered $\alpha$-MoO$_3$ within heterostructure during lithiation. To further identify the phase transition during the lithiation/delithiation of these Mo-based electrodes, we carried out differential capacity analysis (dQ/dV) by numerical differentiation of data in Fig. 4a−c[29,30]. As shown in Supplementary Fig. 10, the features at ~2.8 V vs. Li/Li$^+$ in dQ/dV plot of $\alpha$-MoO$_3$ are absent in its 5th cycle, suggesting the irreversible phase transition of the $\alpha$-MoO$_3$ electrode. In sharp contrast, no irreversible phase transition is found in dQ/dV plot of $\alpha$-MoO$_3$/$\eta$-Mo$_4$O$_{11}$ electrode, which maintains the pristine redox peaks well in its 5th cycle.

Typical charge/discharge curves of the $\alpha$-MoO$_3$, $\alpha$-MoO$_3$/$\eta$-Mo$_4$O$_{11}$, and $\eta$-Mo$_4$O$_{11}$ electrodes at specific current from 0.1 A g$^{-1}$ to 5 A g$^{-1}$ are shown in Supplementary Fig. 11, and the corresponding rate capabilities of the three electrodes are compared in Fig. 4d. Layered $\alpha$-MoO$_3$, possessing a large interlayer distance, could favor fast Li$^+$ diffusion through the two-dimensional diffusion channels. The rate performance of $\alpha$-MoO$_3$, however, is compromised by its irreversible phase transition and fast structural degradation, which slow down its electrode kinetics. When $\alpha$-MoO$_3$ is stabilized in the heterostructure, the $\alpha$-MoO$_3$/$\eta$-Mo$_4$O$_{11}$ electrode exhibits both large specific capacity and high rate performance. At a high specific current of 5 A g$^{-1}$, the corresponding capacity retention of the $\alpha$-MoO$_3$/$\eta$-Mo$_4$O$_{11}$ electrode (with reference to the capacity at 0.1 A g$^{-1}$) is ~41%, which is larger than ~31% of $\alpha$-MoO$_3$ and ~16% of $\eta$-Mo$_4$O$_{11}$.

The electrochemical impedance spectroscopy (EIS) spectra of the three electrodes were measured at open-circuit potentials and are shown in Supplementary Fig. 12. It can be observed that the charge transfer resistance of the $\alpha$-MoO$_3$/$\eta$-Mo$_4$O$_{11}$ electrode calculated by fitted EIS is the smallest among the three electrodes, as shown in Supplementary Fig. 12 and Supplementary Table 3, further demonstrating the improved charge transfer efficiency for the heterostructure. Meanwhile, the chemical diffusion coefficient of lithium ($D_{Li^+}$) of $\alpha$-MoO$_3$, $\eta$-Mo$_4$O$_{11}$, and $\alpha$-MoO$_3$/$\eta$-Mo$_4$O$_{11}$ electrodes were calculated (Supplementary Fig. 13) by conducting the galvanostatic intermittent titration (GITT) technique during the lithiation process. The $D_{Li^+}$ values in $\alpha$-MoO$_3$ electrode are substantially higher than those in $\eta$-Mo$_4$O$_{11}$ electrode owing to the large ionic diffusion channels. However, the irreversible phase transition results in the dramatic decrease of the diffusion coefficient of Li$^+$ in $\alpha$-MoO$_3$ electrode. Compared with both $\alpha$-MoO$_3$ and $\eta$-Mo$_4$O$_{11}$ electrodes, $\alpha$-MoO$_3$/$\eta$-Mo$_4$O$_{11}$ electrode possesses fast and stable Li$^+$ diffusion, which could be assigned to the solid-state interfacial chemistry and improved structural stability.

Based on the ultraviolet photoelectron spectroscopy (UPS) and ultraviolet-visible absorption spectroscopy measurements (Supplementary Fig. 14a), the schematic energy-level diagrams of $\alpha$-MoO$_3$, $\eta$-Mo$_4$O$_{11}$, and heterostructured $\alpha$-MoO$_3$/$\eta$-Mo$_4$O$_{11}$ were proposed as shown in Supplementary Fig. 14b. The Fermi levels of both $\alpha$-MoO$_3$ and $\eta$-Mo$_4$O$_{11}$ are all located near their Conduction Bands, suggesting their n-type semiconductor feature. As $\alpha$-MoO$_3$ exhibits smaller work function (4.28 eV) as compared to that of $\eta$-Mo$_4$O$_{11}$ (4.58 eV), electrons tend to transfer from $\alpha$-MoO$_3$ to $\eta$-Mo$_4$O$_{11}$ across the interface. The electron flow leads to the accumulation of positive charge on the $\alpha$-MoO$_3$ side and negative charge on the $\eta$-Mo$_4$O$_{11}$ side near the interface. Simultaneously, the energy levels of $\alpha$-MoO$_3$ shift upward whereas those of $\eta$-Mo$_4$O$_{11}$ bend downward near the interface until their Fermi levels reach equilibrium[31]. Thus, a built-in electric field with a direction pointing from $\alpha$-MoO$_3$ to $\eta$-Mo$_4$O$_{11}$ is formed at the $\alpha$-MoO$_3$/$\eta$-Mo$_4$O$_{11}$

heterojunction. Given that $\eta$-Mo$_4$O$_{11}$ domains are randomly embedded within the $\alpha$-MoO$_3$ matrix, Li$^+$ ions need to be transferred from $\alpha$-MoO$_3$ to $\eta$-Mo$_4$O$_{11}$ during the discharge process. During the discharge process, the as-formed built-in electric field can accelerate Li$^+$ diffusion from $\alpha$-MoO$_3$ to $\eta$-Mo$_4$O$_{11}$, thus improving the electrode kinetics for the discharge process. Under the built-in electric field, Li$^+$ could accumulate at $\eta$-Mo$_4$O$_{11}$ side to neutralize the negative charges, and the electric field around the hetero-interface may finally vanish after the charge balance[32]. Upon the charge process, Li$^+$ ions are first extracted from $\alpha$-MoO$_3$ owing to its direct contact with electrolyte. As $\alpha$-MoO$_3$ possesses a more open structure for Li$^+$ diffusion, the fast leaching of Li$^+$ in $\alpha$-MoO$_3$ will lead to a Li$^+$ concentration gradient across the $\alpha$-MoO$_3$/$\eta$-Mo$_4$O$_{11}$ interface, thus generating a reversed built-in electric field during the charge process. Under such an electric field, the transfer of Li$^+$ from $\eta$-Mo$_4$O$_{11}$ to $\alpha$-MoO$_3$ can be accelerated, improving the electrode kinetics for charge process[33]. Overall, the constructed $\alpha$-MoO$_3$/$\eta$-Mo$_4$O$_{11}$ hetero-interface is beneficial to improve Li$^+$ intercalation and deintercalation in this heterostructured electrode during both charge and discharge processes.

The structural stabilities of different electrodes were further investigated by cycling tests. Figure 4e compares the cycle performances of the three electrodes between 1.0 and 3.5 V (vs. Li/Li$^+$) at a specific current of 0.5 A g$^{-1}$. It is clear that the $\alpha$-MoO$_3$ electrode exhibits fast capacity fading with only approximately 27% capacity retained after 300 cycles, suggesting continuous structural degradation during cycling. In comparison, the $\eta$-Mo$_4$O$_{11}$ electrode presents superior structural stability with about 82% capacity retained after 300 cycles, demonstrating that it can be used as good structural stabilizer for the heterostructure design. Nevertheless, the specific capacity of the $\eta$-Mo$_4$O$_{11}$ electrode at 0.5 A g$^{-1}$ is the lowest among the three electrodes due to its poor rate capability. Remarkably, the $\alpha$-MoO$_3$/$\eta$-Mo$_4$O$_{11}$ electrode displays both the highest specific capacity and the best cycle performance with a negligible capacity loss after 300 cycles. The superior structural stability of the heterostructure can be further demonstrated by a long-term cycling test. When cycled at a low specific current of 0.5 A g$^{-1}$, the $\alpha$-MoO$_3$/$\eta$-Mo$_4$O$_{11}$ electrode can retain over 90% of the initial capacity after 1000 cycles. In addition, as depicted in Supplementary Fig. 15a, b, the redox peaks of $\alpha$-MoO$_3$/$\eta$-Mo$_4$O$_{11}$ electrode are well retained after 1000 cycles and the corresponding dQ/dV plots at the 1st and 1000th cycles are nearly overlapped with minimal capacity loss, further demonstrating the high stability of the heterostructure with coherent interface. In sharp contrast, the $\alpha$-MoO$_3$ displays an irreversible capacity loss of ~73% after 300 cycles at a specific current of 0.5 A g$^{-1}$. Moreover, the redox peaks in dQ/dV plot of $\alpha$-MoO$_3$ are remarkably reduced and new peaks emerge even after 300 cycles, suggesting severe structural change.

As the $\alpha$-MoO$_3$/$\eta$-Mo$_4$O$_{11}$ electrode was subjected to prolonged cycling at 2 A g$^{-1}$ (Fig. 4f), the initial reversible capacity is ~170 mAh g$^{-1}$ and about 81% of the initial capacity can be retained after 3000 cycles, substantially outperforming previously reported $\alpha$-MoO$_3$-based positive electrodes (Supplementary Table 4), including H$_2$O molecule modified $\alpha$-MoO$_3$, oxygen-deficient $\alpha$-MoO$_{3-x}$, $\alpha$-MoO$_{3-x}$/carbon nanotube, and pre-intercalated $\alpha$-MoO$_3$ (Supplementary Fig. 16)[10,13,15,23,34,35]. Importantly, the common phenomenon of fast capacity decay during initial cycles for the $\alpha$-MoO$_3$-based electrode as reported in literature is not observed in the present study for the $\alpha$-MoO$_3$/$\eta$-Mo$_4$O$_{11}$ heterostructured electrode, suggesting greatly enhanced structural stability of $\alpha$-MoO$_3$ within the heterostructure. Supplementary Fig. 17a−d presents the FESEM images of the $\alpha$-MoO$_3$ and $\alpha$-MoO$_3$/$\eta$-Mo$_4$O$_{11}$ thin films collected after different cycle numbers at 0.5 A g$^{-1}$, respectively. It is noted that the morphology of the $\alpha$-MoO$_3$/$\eta$-Mo$_4$O$_{11}$ thin film was well maintained after 1000 cycles; however, severe morphology changes along with nanoparticles formation and aggregation, and cavities formation were observed in the $\alpha$-MoO$_3$ thin film after only 5 cycles (Supplementary Fig. 18). The photo images of different electrodes after

300 cycles in Supplementary Fig. 19 visually confirm the excellent stability of the α-MoO₃/η-Mo₄O₁₁ thin film during the cycling process. In strong contrast, drastic exfoliation of the active material from the α-MoO₃ thin film electrode was detected after 300 cycles, indicating large volume change and electrode pulverization occurred during the cycling.

## Lattice pinning via coherent interface

To grasp insight into the mechanism of the excellent structural stability of the α-MoO₃/η-Mo₄O₁₁ heterostructure, we have conducted ex situ XRD measurements on different electrodes at various charge and discharge states to elucidate the potential-dependent structure evolution and lattice variation (Fig. 5a, b). For α-MoO₃, the gradual disappearance of the (020) peak belonging to the pristine α-MoO₃ along with the emerging (030) peak of the lithiated α-MoO₃ ($Li_xMoO_3$, x~0.25), at ~2.8 V vs. Li/Li⁺, confirms the irreversible phase transformation during the initial discharge process[10,13,14]. Simultaneously, the intensive (110) and (040) peaks ascribed to α-MoO₃ vanish and a new broad peak appears in the following lithiation process, indicating the original stacking order of Mo−O octahedron layers becomes strongly disordered[28]. In contrast to α-MoO₃, all of the characteristic peaks for the α-MoO₃/η-Mo₄O₁₁ heterostructure are well retained with only reversible peak shift during charge/discharge, demonstrating α-MoO₃ layered structure can be stabilized without phase transition in the heterostructure. Meanwhile, the corresponding interlayered spacing of (020) planes of α-MoO₃ in heterostructure increases from 6.80 Å at 3.5 V to 6.96 Å at 1.0 V (expansion rate ~2%), whereas the initial interlayer spacing of the pure α-MoO₃ exhibits a considerably increased expansion rate of ~16% (from 6.80 to 7.90 Å), demonstrating the coherent interface can effectively pin lattice expansion of α-MoO₃ in heterostructure. Unlike α-MoO₃, the monoclinic η-Mo₄O₁₁ presents minimal and reversible lattice change during both lithiation and delithiation processes (Supplementary Fig. 20), suggesting that it could be the perfect structural stabilizer to constrain lattice variation of α-MoO₃. It should be noted that the coherent interface not only suppresses the lattice variation of α-MoO₃ but also further reduces the lattice variation of η-Mo₄O₁₁ in the heterostructure. As compared between Fig. 5b and Supplementary Fig. 20, the diffraction peak shift of η-Mo₄O₁₁ in heterostructure during charge and discharge is even smaller than that of the pure η-Mo₄O₁₁.

The ex situ XPS measurements were performed to investigate the electrochemical reversibility of the α-MoO₃/η-Mo₄O₁₁ heterostructure (Fig. 5c). When fully delithiated at 3.5 V, the α-MoO₃/η-Mo₄O₁₁ heterostructure contains only Mo⁶⁺ and Mo⁵⁺, whereas the pure α-MoO₃ presents ~18% Mo⁴⁺ in addition to Mo⁶⁺ and Mo⁵⁺, which suggests that more inactive Li⁺ are trapped in the pure α-MoO₃ associated with the initial irreversible phase transition. Another interesting feature is that the fraction of Mo⁴⁺ at the full lithiation state in α-MoO₃/η-Mo₄O₁₁ (~52%) is obviously higher than that in α-MoO₃ (~41%; Supplementary Fig. 21), suggesting that the enhancement in specific capacity of the heterostructure originates from the enhanced redox activity of Mo⁶⁺ to Mo⁴⁺. The atomic structure of the cycled α-MoO₃/η-Mo₄O₁₁ sample, which underwent 100 cycles at a specific current of 0.5 A g⁻¹, was further analyzed by HADDF-STEM. As seen in Supplementary Fig. 22, both α-MoO₃ and η-Mo₄O₁₁ phases connected by the coherent interface can be clearly observed. The Mo atomic arrangements in the red square agree well with the orthorhombic α-MoO₃ viewed along [100] zone axis, while the Mo atomic arrangements in the yellow square agree well with η-Mo₄O₁₁ viewed along [001] zone axis, demonstrating that the coherent interface between α-MoO₃ and η-Mo₄O₁₁ is highly stable during the cycling process. Schematic Fig. 5d illustrates that the origin of the irreversible phase transition and structure degradation in α-MoO₃ lie in the large lattice expansion during lithiation process. In sharp contrast, as illustrated in Fig. 5e, the epitaxial lattice of α-MoO₃

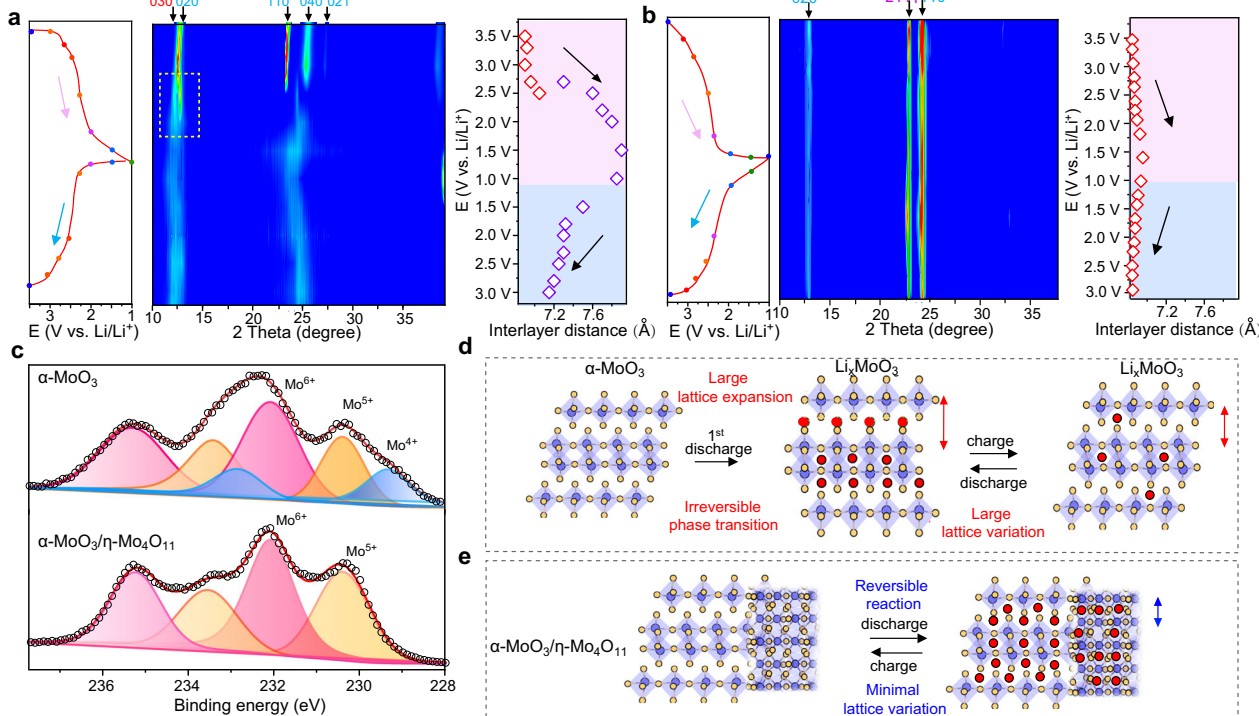

**Fig. 5 | Structure evolution upon Li⁺ intercalation/deintercalation. a, b** Ex situ XRD measurements of the α-MoO₃ and α-MoO₃/η-Mo₄O₁₁ electrodes during the first discharge−charge processes in a voltage range between 1.0 and 3.5 V (vs. Li/Li⁺). η respresnts η-Mo₄O₁₁. **c** Mo 3*d* XPS spectra of α-MoO₃ and α-MoO₃/η-Mo₄O₁₁ at the delithiated state (3.5 V vs. Li/Li⁺). **d, e** Mechanism of structural degradation in α-MoO₃ and enhanced structural stability of α-MoO₃ induced by lattice pinning in α-MoO₃/η-Mo₄O₁₁ during lithiation/delithiation processes. The purple spheres represent Mo atoms, the yellow spheres represent oxygen atoms, and red spheres represent lithium atoms.

is well constrained by the coherent interface due to the strong ionic bonds between α-MoO₃ and η-Mo₄O₁₁. With the strong lattice pinning effect, the layered structure of α-MoO₃ is perfectly stabilized in the heterostructure, allowing efficient Li⁺ intercalation and deintercalation with minimal lattice variation.

## Insights from theoretical calculations

DFT calculations were performed to gain in-depth insights into the improved structure stability of layered α-MoO₃ induced by the lattice pinning effect. f-Li$_{0.25}$MoO₃ was used as a prototype for the lithiated state to compare with the pristine α-MoO₃ because the irreversible phase transition occurs at this lithiation state of Li$_x$MoO₃ ($x = 0.25$). A hypothetical c-Li$_{0.25}$MoO₃ structure with fixed interlayered spacing along the $b$ axis as the pristine α-MoO₃ was used to simulate the pinning effect on the suppression of lattice expansion. Supplementary Fig. 23a–c shows the relaxed lattice structures of α-MoO₃, f-Li$_{0.25}$MoO₃,

and c-Li$_{0.25}$MoO₃, respectively. An increased interlayer spacing of 7.9 Å along the $b$ axis in f-Li$_{0.25}$MoO₃ was observed as compared to 6.8 Å in α-MoO₃, which is in good agreement with the experimental result. Simultaneously, the distorted Mo⁶⁺ ions in MoO₆ octahedra of α-MoO₃ was found to move to the octahedral center after lithiation (Fig. 6a–c). It is well known that the off-center displacement of d⁰ Mo in MoO₆ octahedra of α-MoO₃ (Mo⁶⁺) was ascribed to the second-order Jahn–Teller (SOJT) effect through the interaction between the empty d-orbitals (d⁰) of Mo cations and filled p-orbitals of oxygen ions[36–39]. The formation of Mo⁵⁺/Mo⁴⁺ associated with Li⁺ intercalation into the layered α-MoO₃, however, breaks this distortion state, resulting in the migration of Mo ions back to the center of MoO₆ octahedra[40]. Therefore, lithiation in pristine α-MoO₃ induces large lattice expansion together with Mo migration, which could promote the irreversible phase transition with poor structural stability (Supplementary Fig. 24).

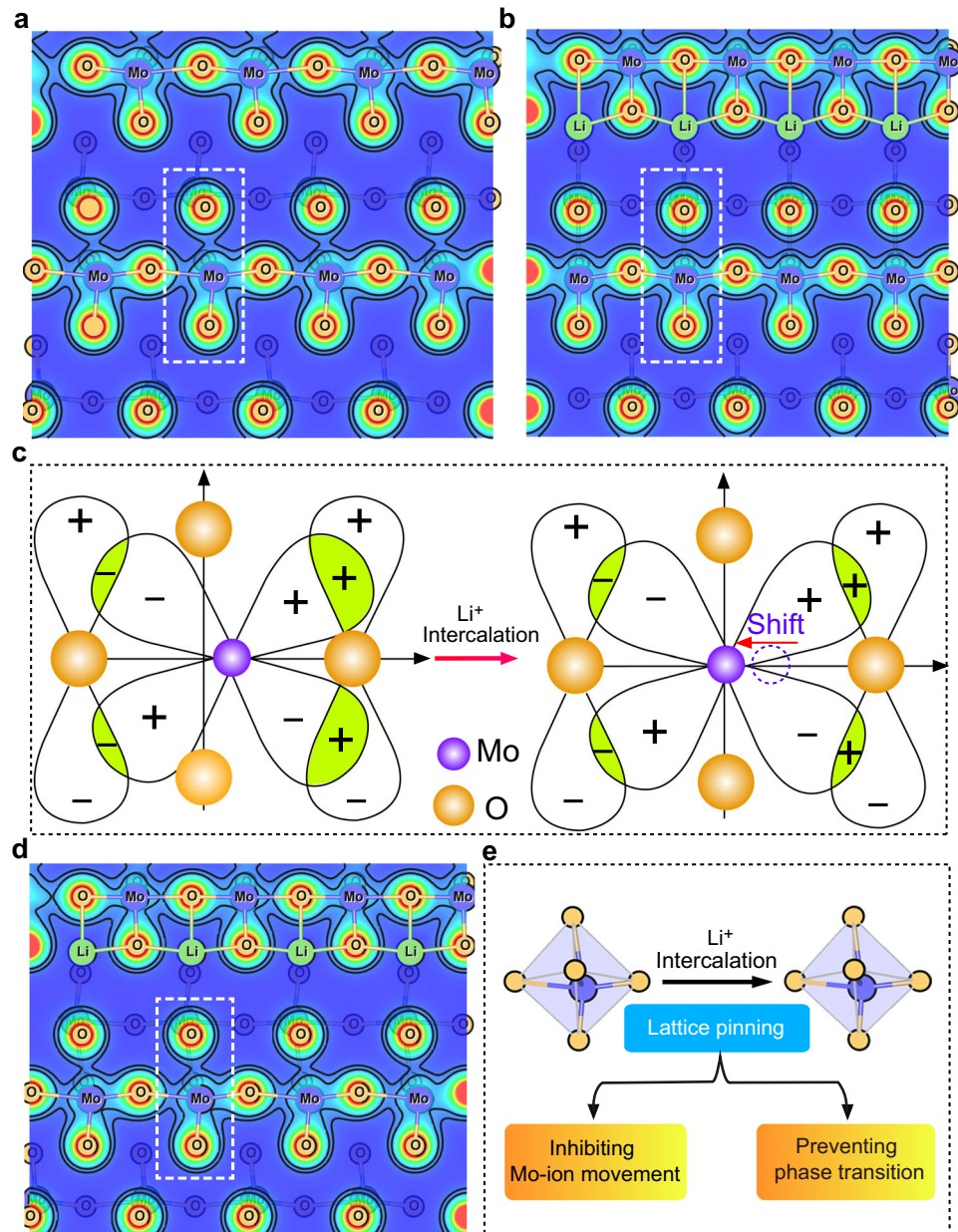

**Fig. 6 | Insights from theoretical calculations. a, b** Differential charge density of MoO₃ and f-Li$_{0.25}$MoO₃. The blue and red regions represent the depletion and accumulation of charge, respectively. **c** Layered MoO₃ with Mo off-centering and the displacement of Mo ion being shifted toward the orbital center during lithiation. **d** Differential charge density of c-Li$_{0.25}$MoO₃. **e** α-MoO₃ with constrained lattice eliminates the migration of Mo ions and phase transition during lithiation.

Interestingly, the Mo ions in $MoO_6$ octahedra of $c$-$Li_{0.25}MoO_3$ retain the same position as the pristine $\alpha$-$MoO_3$ (Fig. 6d), suggesting the migration of Mo ions is driven by lattice expansion during lithiation. By suppressing the interlayer expansion along the $b$ direction, the migration barrier of Mo ions is greatly enhanced during the discharge process, which effectively inhibits the displacement of Mo ions to the center of $MoO_6$ octahedra. In addition, partial density of states (PDOS) analysis indicates enhanced energy for the occupied Mo/O states in $f$-$Li_{0.25}MoO_3$ as compared to that in $c$-$Li_{0.25}MoO_3$, demonstrating weakened Mo−O interaction stemming from increased interlayered spacing in $f$-$Li_{0.25}MoO_3$ (Supplementary Fig. 25). As indicated in Supplementary Fig. 26, there are three types of oxygen atoms, i.e., $O_t$, $O_a$ and $O_s$, in the pristine $\alpha$-$MoO_3$. It can be seen that the Mo−$O_t$ bonds in $f$-$Li_{0.25}MoO_3$ elongate from 2.372 to 2.461 Å, while the length of Mo−$O_t$ bond in $c$-$Li_{0.25}MoO_3$ only changes from 2.372 to 2.371 Å (Supplementary Table 5) after lithiation. The notably increased Mo−$O_t$ bonds greatly increase the probability for Mo migration in $\alpha$-$MoO_3$, thus enabling the irreversible phase transition during the initial lithiation. Based on the DFT calculations, it is speculated that lattice pinning via the coherent interface can significantly increase the structural stability of $\alpha$-$MoO_3$ in the heterostructure, successfully inhibiting the adverse phase transition (Fig. 6e).

### All-solid-state thin-film microbattery

Owing to the solid−solid electrode /electrolyte interface in all-solid-state lithium batteries, the positive electrode materials with large volume change during charge/discharge are detrimental to interface stability, which could result in fast capacity fading and even battery failure[41]. Herein, the heterostructured $\alpha$-$MoO_3$/$\eta$-$Mo_4O_{11}$ electrode with minimal lattice variation during cycling is anticipated to find desirable application in solid-state lithium batteries[42,43]. To identify the compatibility of the $\alpha$-$MoO_3$/$\eta$-$Mo_4O_{11}$ electrode with solid electrolyte, we fabricated an all-solid-state thin-film lithium microbattery using $\alpha$-$MoO_3$/$\eta$-$Mo_4O_{11}$ as the positive electrode, LiPON as the solid electrolyte, and metal lithium as the negative electrode (Fig. 7a and Supplementary Fig. 27). The suitable ionic conductivity ($3 \times 10^{-6}$ S cm$^{-1}$; $298 \pm 2$ K) and low electric conductivity ($1 \times 10^{-10}$ S cm$^{-1}$; $298 \pm 2$ K) of LiPON with high stability make it an appropriate solid electrolyte for solid-state thin-film lithium batteries[19,35,44]. The LiPON was sputtered using a $Li_3PO_4$ target under $N_2$ atmosphere at 298 K. During the RF magnetron sputtering process, the introduced $N_2$ gas was split into nitrogen ions and electrons under the high electric field, resulting in a reactive nitrogen cation species. The ionized N species can react with the Li-P-O species to form LiPON[45,46]. As shown in the cross-section FESEM image of the solid-state battery (Fig. 7b), the LiPON affords close contact with both the $\alpha$-$MoO_3$/$\eta$-$Mo_4O_{11}$ electrode and lithium metal electrode, guaranteeing efficient charge transfer at the interface with low interfacial impedance. In addition, the STEM image and corresponding energy-dispersive X-ray spectroscopy (EDS) elemental mappings are shown in Supplementary Fig. 28 and Fig. 7c, respectively. Well-separated and uniform distributions of Mo and P can be clearly observed in the sample, further demonstrating the intimate interface contact.

The charge/discharge curves of the solid-state thin-film lithium microbattery based on the $\alpha$-$MoO_3$/$\eta$-$Mo_4O_{11}$ positive electrode at different specific currents are shown in Supplementary Fig. 29. At a low specific current of 0.05 A g$^{-1}$, the microbattery delivers a large capacity of ~300 mAh g$^{-1}$ (~0.075 mAh cm$^{-2}$) at the first cycle (Fig. 7d). Even at a high specific current of 2 A g$^{-1}$, it can still exhibit ~90 mAh g$^{-1}$, suggesting good rate performance. The specific capacity and rate performance of the $\alpha$-$MoO_3$/$\eta$-$Mo_4O_{11}$ electrode in the solid-state thin-film battery are relatively lower than those of the $\alpha$-$MoO_3$/$\eta$-$Mo_4O_{11}$ electrode tested in organic electrolyte, suggesting limited charge transport kinetics in solid-state battery configuration. Nevertheless, the as-fabricated solid-state microbattery device is highly flexible and

robust (Fig. 7e). The cycling test of the device showed negligible capacity change under various bending angles from 0 to 180°. Even after cycling at different bending states, the interface between $\alpha$-$MoO_3$/$\eta$-$Mo_4O_{11}$ electrode and LiPON electrolyte maintains the intimate contact, suggesting the good flexibility of the solid-state microbattery and potential application as power sources for various flexible electronics (Supplementary Figs. 30 and 31). Importantly, Fig. 7f shows the long-term cycling performance of the solid-state microbattery, presenting ~74% capacity retention even after 4000 cycles at 2 A g$^{-1}$. The long-term cycling performance of the $\alpha$-$MoO_3$/$\eta$-$Mo_4O_{11}$ electrode within a solid-state microbattery is well-placed among previously reported solid-state microbatteries (Supplementary Fig. 32)[19,35,45], demonstrating the substantially enhanced battery cycle life in solid-state configuration when the $\alpha$-$MoO_3$/$\eta$-$Mo_4O_{11}$ heterostructure is employed.

Based on the Galvanostatic charge/discharge profile in Supplementary Fig. 29, the $\alpha$-$MoO_3$/$\eta$-$Mo_4O_{11}$-based thin film micorbattery delivers a specific energy density of ~0.48 mWh with an areal energy density of 0.16 mWh cm$^{-2}$. Moreover, the $\alpha$-$MoO_3$/$\eta$-$Mo_4O_{11}$ based solid-state microbattery can achieve superior electrochemical performance in comparison with the $LiCoO_2$ and $LiMn_2O_4$-based solid-state microbatteries, especially in specific capacity, max energy density, and cycle life[19,44]. Importantly, for the $\alpha$-$MoO_3$/$\eta$-$Mo_4O_{11}$ and $\alpha$-$MoO_3$ thin film deposition, a cheap Mo metal target can be directly used in sputtering, making the fabrication cost much lower than $LiCoO_2$ and $LiMn_2O_4$ thin films. In addition, in contrast to traditional Li-contained positive electrode thin films ($LiCoO_2$ and $LiMn_2O_4$) that necessitate a high annealing temperature around 973 K, the $\alpha$-$MoO_3$/$\eta$-$Mo_4O_{11}$ as well as $\alpha$-$MoO_3$ thin films can be well crystallized at a much lower annealing temperature of 673 K, making battery on-chip integration compatible without damaging the chip.

## Discussion

In summary, we demonstrate a lattice pinning strategy via constructing coherent interface to stabilize $\alpha$-$MoO_3$ in the intergrown $\alpha$-$MoO_3$/$\eta$-$Mo_4O_{11}$ heterostructure. This strategy boosts Li$^+$ storage with substantially prolonged cycle life and improved rate capability. With the rational heterostructure design, the epitaxial lattice of $\alpha$-$MoO_3$ is effectively constrained by $\eta$-$Mo_4O_{11}$ at the coherent interface via the strong ionic bonds, which successfully suppresses lattice expansion along $b$ direction and eliminates the adverse phase transition of $\alpha$-$MoO_3$ during the lithiation process. The DFT calculations indicate that lithiation-induced interlayer expansion in $\alpha$-$MoO_3$ weakens the Mo−O bonds and allows Mo migration in $MoO_6$ octahedra, which could drive the irreversible phase transition to form the unstable $Li_xMoO_3$ phase. With minimal lattice variation during lithiation/delithiation, the $\alpha$-$MoO_3$/$\eta$-$Mo_4O_{11}$ heterostructure affords substantially extended cycle life without the fast capacity loss within initial cycles as compared to individual $\alpha$-$MoO_3$. The high structural stability associated with minimal lattice variation of the $\alpha$-$MoO_3$/$\eta$-$Mo_4O_{11}$ heterostructure is further demonstrated in all-solid-state thin-film lithium microbattery, which presents a long-term cycling stability with about 74% capacity retention after 4000 cycles at 2 A g$^{-1}$. This study demonstrates effective lattice pinning in layered $\alpha$-$MoO_3$ via coherent interface design, and provides new insights to design highly stabilized intercalation hosts for sustainable batteries.

## Methods

### Materials synthesis

The Mo target is industrially available material, which was purchased from Wuxi Kaistar Electro-optic Material Co., LTD. The $\alpha$-$MoO_3$, $\alpha$-$MoO_3$/$\eta$-$Mo_4O_{11}$, and $\eta$-$Mo_4O_{11}$ thin films were deposited on Pt/Ti/glass or Pt/Ti/stainless steel substrates using a pure Mo target (75 mm in diameter and 5 mm in thickness) through DC magnetron sputtering (SKY Technology Development Co., Ltd, China) under $O_2$ and Ar

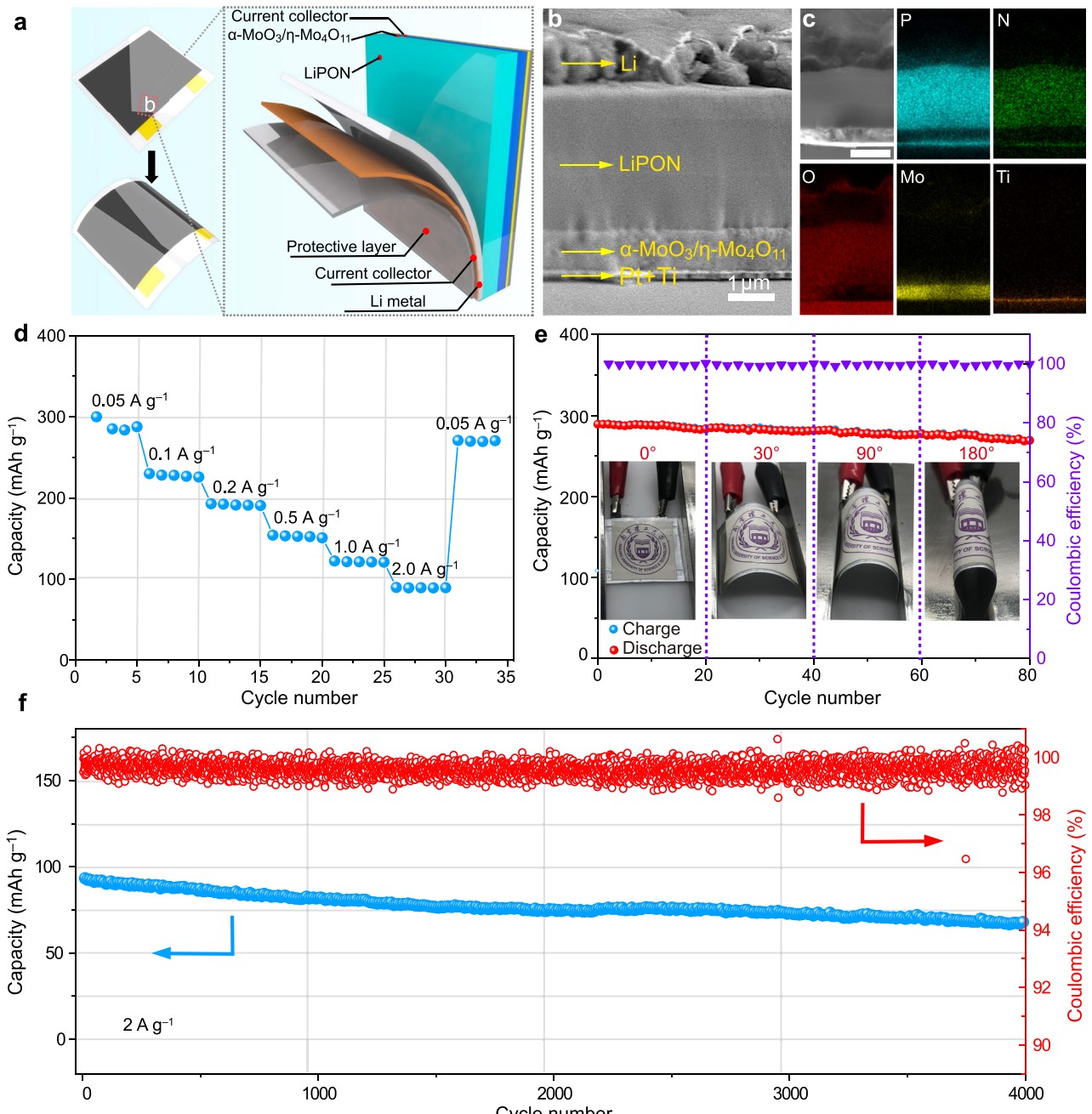

**Fig. 7 | All-solid-state thin-film lithium microbattery. a** Schematic diagram illustrating the configuration of all-solid-state thin-film lithium microbattery. **b** The cross-section FESEM image of the thin-film microbattery. **c** The cross-section STEM image and corresponding EDS elemental mappings of the thin-film microbattery. **d** Rate performance of the thin-film microbattery. **e** Cycle performance of the flexible thin-film microbattery at different bending states at 0.05 A g⁻¹. **f** Long-term cycle performance of thin-film microbattery at 2 A g⁻¹. The solid electrochemical energy storage tests were carried out at 298 ± 2 K.

atmosphere. Before deposition, a background pressure less than $1 \times 10^{-5}$ Pa was first reached in the chamber. The distance between target and substrate was kept at 10 cm and the substrate rotation is 20 rpm. The DC power was fixed at 60 W. The $\alpha\text{-MoO}_3$ thin film was deposited at 573 K, in a working pressure of ~1 Pa introduced by mass flow controllers with 40.0 sccm Ar and 10.0 sccm $O_2$ (25% $O_2$/Ar ratio). $\alpha\text{-MoO}_3/\eta\text{-Mo}_4O_{11}$ thin film was deposited at 573 K, in a working pressure of ~1 Pa introduced by mass flow controllers with 40.0 sccm Ar and 5.0 sccm $O_2$ (13% $O_2$/Ar ratio). $\eta\text{-Mo}_4O_{11}$ thin film was deposited at 573 K, in a working pressure of ~1 Pa introduced by mass flow controllers with 40.0 sccm Ar and 3.0 sccm $O_2$ (8% $O_2$/Ar ratio). All the films were deposited for 1.5 h and then annealed in the chamber at 673 K for 1 h to

obtain the final $\alpha\text{-MoO}_3$, $\alpha\text{-MoO}_3/\eta\text{-Mo}_4O_{11}$, and $\eta\text{-Mo}_4O_{11}$ thin films. The average thickness of $\alpha\text{-MoO}_3$, $\alpha\text{-MoO}_3/\eta\text{-Mo}_4O_{11}$, and $\eta\text{-Mo}_4O_{11}$ thin films are about 1.3, 1.0, and 1.1 μm, respectively. The mass loadings of the $\alpha\text{-MoO}_3$, $\alpha\text{-MoO}_3/\eta\text{-Mo}_4O_{11}$, and $\eta\text{-Mo}_4O_{11}$ thin films determined by a Sartorius Analytical Balance (CPA225D, with resolution of 10 μg) were 0.21, 0.25, and 0.30 mg cm⁻², respectively.

## Materials characterizations
The crystallographic information of the samples was characterized by XRD (Bruker) with Cu Kα radiation, Raman spectroscopy (Jobin-Yvon T6400 Micro-Raman system), XPS (Phi Quantera SXM spectrometer equipped with an Al Kα X-ray excitation source). For the Rietveld

refinement process with the materials analysis using diffraction (MAUD) software, data is first refined according to the background and instrument parameters. Then, we used the structural model of α-MoO$_3$ ($a = 3.95$ Å, $b = 13.83$ Å, $c = 3.69$ Å, $Z = 4$, space group $Pbnm$, ICSD no. 36167) as the starting model in the refinement. Moreover, "excess" peaks that cannot be indexed to α-MoO$_3$ (ICSD no. 36167) were further refined using the structural model of η-Mo$_4$O$_{11}$ ($a = 24.54$ Å, $b = 5.44$ Å, $c = 6.70$ Å, $Z = 4$, space group $P2_1/a$, ICSD no. 15814), generating a low-reliability factor $R_{wp}$. Specifically, the quantitative value (64/36) of weight fraction between α-MoO$_3$ and η-Mo$_4$O$_{11}$ is extracted from the MAUD based on the following equation (Eq. (1))[47,48]:

$$W_i = \frac{S_i(ZMV)_i}{\sum_{j=1}^{N} S_j(ZMV)_j} \tag{1}$$

where $V$, $M$, $Z$, and $S$ are the unit-cell volume of phase $i$, the molecular weight, the number of molecules per unit cell, and the scale factor in the α-MoO$_3$/η-Mo$_4$O$_{11}$ sample, respectively. Considering the thin film texture effect, Rietveld refinement on thin film XRD patterns was carried out by using the harmonic texture model implemented into the MAUD software[49–51]. For the electronic structure, we used UPS (Thermo ESCALAB 250XI), which provides valence band and work function position. The morphologies and microstructures of the samples were investigated by FESEM (FEI Quanta 250 F) and HRTEM (FEI Tecnai G2). A JEM-ARM200F STEM fitted with a double-aberration corrector for both probe-forming and imaging lenses was utilized to perform the HAADF/ABF imaging and EELS; the electron microscope was operated at 200 kV. The attainable resolution of the probe defined by the objective pre-field was 78 pm.

### Electrochemical measurements
Li coin cells using the α-MoO$_3$, α-MoO$_3$/η-Mo$_4$O$_{11}$, and η-Mo$_4$O$_{11}$ thin films as the positive electrodes, 1 M LiClO$_4$ in ethylene carbonate and dimethyl carbonate (EC: DMC, 1:1, v/v) solution as the electrolyte (~100 µL), Celgard 2400 as the separator (16 mm in diameter and 25 µm in thickness), and lithium foils (99.9% in purity) as negative electrodes were assembled into R2025-type coin cells in an argon-filled glove box (O$_2$ < 0.1 ppm, H$_2$O < 0.1 ppm). The porosity and average pore size of the separator are 41%, and 0.043 µm, respectively. Moreover, the lithium foil has a diameter of 14.5 mm and a thickness of 500 µm.

The $D_{Li^+}$ is calculated by GITT according to the following equation (Eq. (2)):

$$D_{GITT} = \frac{4}{\pi\tau}\left(\frac{n_m V_m}{S}\right)^2 \left(\frac{\Delta E_s}{\Delta E_t}\right)^2 \tag{2}$$

where $n_m$ represents the mole value, $\tau$ is relaxation time, and $S$ represents the area of the electrode, $V_m$ is the mole volume, $\Delta Et$ represents the changed voltage during the discharge, and $\Delta Es$ is the voltage response under the pulse current.

The all-solid-state thin-film battery was fabricated through a layer-by-layer deposition procedure. Stainless steel foil with a Ti/Pt transition layer was used as the substrate, and the α-MoO$_3$/η-Mo$_4$O$_{11}$ thin film positive electrode was first deposited by DC sputtering. The LiPON solid electrolyte film and lithium metal film were then sequentially deposited onto the α-MoO$_3$/η-Mo$_4$O$_{11}$ thin film (Supplementary Fig. 27). Typically, the LiPON electrolyte film with a thickness of 2.5 µm was deposited onto the positive electrode by RF reactive magnetron sputtering (SKY Technology Development Co., Ltd, China) using a Li$_3$PO$_4$ target (75 mm in diameter and 5 mm in thickness) under N$_2$ atmosphere at 298 K for 15 h. Then, the lithium metal layer of approximately 2.5 µm was deposited on the LiPON layer by thermal evaporation (SKY Technology Development Co., Ltd, China). It should be noted that the area of LiPON is larger than those of the positive electrode and negative electrode to avoid a short circuit at the edge. After the deposition of 500 nm Cu current collector by RF reactive magnetron sputtering (SKY Technology Development Co., Ltd, China) using a Cu target, the all-solid-state thin-film microbattery was finally sealed with aluminum-plastic film. Galvanostatic charge–discharge measurements were operated in the voltage range between 1.0 and 3.5 V (vs. Li/Li$^+$) at different specific currents using a Neware BTS4000 battery test system. CV measurements were conducted on an electrochemical workstation (Biologic VSP). EIS measurements were carried out by applying a potentiostatic amplitude of 10 mV over the frequency range of 100.0 kHz to 0.1 Hz on the Biologic VSP. Six data points are collected per decade of frequency. The open-circuit voltage time applied before carrying out the EIS measurement is ~10 h. Zview software is used to fit the EIS and calculate the errors between the raw and fitted data. The electrochemical energy storage tests were carried out at 298 ± 2 K.

### DFT calculation
First-principles calculations based on DFT were performed to understand the effect of Li-ion intercalation on the crystalline structure of α-MoO$_3$ using the Vienna Ab initio Simulation Package (VASP). The projector-augmented wave (PAW) method was used to describe the ion-electron interactions. Perdew–Burke–Ernzerhof (PBE) version of generalized gradient approximation (GGA) was adopted for the exchange-correlation energy. A kinetic energy cutoff of 520 eV was used for the plane wave expansion of the valence electron wave functions. A dense Γ-centered Monkhorst-Pack $k$-point mesh with a sampling density of 0.04 Å$^{-1}$, $10^{-6}$ eV/cell in total energy, and $10^{-2}$ eV/Å in force were adopted for the convergence criterion during structural optimization. Because of the layer structure of MoO$_3$, van der Waals density functional (vdWDF) of optB86b-vdW functional was performed during structural optimization. Differential charge density was calculated by the difference between the total valence charge density of the structure and the superposition of the valence charge densities of neutral atoms.

## Data availability
The data that support the findings detailed in this study are available in the Article and its Supplementary Information or from the corresponding authors upon request. Source data are provided with this paper.

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

## Acknowledgements

This work is supported by the National Key R&D Program of China (2020YFB2007400 to H.X.), National Natural Science Foundation of

China (Nos. 52272218 to H.X., 52072179 to T.Z., 52061135201 to H.X., 51972174 to H.X., 22209092 to S.S.), Natural Science Foundation of Jiangsu Province (Nos. BK20200073 to T.Z.), and Fundamental Research Funds for the Central Universities (No. 30920041118 to H.X.).

## Author contributions

H.X. and T.Z. conceived the project. S.S. synthesized the samples, performed the structural characterizations and electrochemical measurements, and analyzed the data. Z.H., X.L. and D.S. performed the STEM measurements and structural analysis. W.L., Q.X. and L.X. performed the EIS, XPS, and Raman measurements. H.X., T.Z. and S.S. wrote the manuscript. All authors analyzed the results and commented on the manuscript.

## Competing interests

The authors declare no competing interests.
