## [Peer Review File · Nature Communications]

nature portfolio

Peer Review FileReviewer comments, first round -

Reviewer #1 (Remarks to the Author):

This is an interesting manuscript demonstrating an attempt to redesign α -MoO₃/ η -Mo₄O₁₁ heterostructure as an alternative to avoid high lattice expansion during electrochemical cycling. Authors claim such heterojunction enhance Li⁺ mobility and storage with longer life cycle and/or specific capacity. Although this is a good work hypothesis, there are critical issues against the publication of this manuscript. While the journal is highly reputed, any misleading data either needs to be corrected, remeasured or clarified.

Taking these facts into consideration, following critical issues are described below:

Authors have attempted to design a solid state interface with α -MoO₃ and isostructural η -Mo₄O₁₁ to reduce large lattice expansion of α -MoO₃ due to Li intercalation or deintercalation. They have focused only on zero-strain interfacial component. However, the solid state chemistry is not so trivial. The charge transfer (including Li intercalation) depends on the specific band positions of two interfacial materials having significant influence on the fermi energy for charge equilibration. Although zero strain material do have an impact on reducing lattice expansion, the mobility of majority carriers (either electrons or holes) across the space charge layers have permanent impact on the Li⁺ extraction or intercalation. These ions have to move following charge equilibration rule. Hence this manuscript has largely excluded the discussion on the solid state interfacial chemistry during charging or discharging.

Figure 2: X-ray diffraction analysis of α -MoO₃. Authors are suggested to revisit the analysis to explain the following ambiguities.

- The review of the data presented in Figure 2c with all the Pbnm space group of the orthorhombic system (α -MoO₃), none of the ICSD data matches with the x-ray diffraction signal intensity presented in the manuscript. The significant mismatch in the intensity of the signals corresponding at two theta 12.9, 23.3, 25.9, 27.4 suggest the wrong assignment of the space group.
- When the XRD patterns are refined, what was the ICSD data (as a cif file used)? Why was JCPDS data (mainly used for the signal positions) given more focus? Authors are advised to give detail description of how they manage to refine this data with such intensity mismatch?
- It was quite difficult to review this data while the intensity figures were removed. If authors want to publish their work in such a reputed journal, the data should be included in the figures or text as it comes from the instrument. Authors are advised to include the intensity data.
- While the signal at 12.9 two theta is characteristic signal for α -MoO₃ (as described in Figure 2c), why is it missing in the α -MoO₃/ η -Mo₄O₁₁ heterostructure [Figure 2(d)]?
- If the pattern is refined using two different phases (α -MoO₃ and η -Mo₄O₁₁) what is the mass content of the two phases?
- Many signals which are supposed to be present for the P21-a of η -Mo₄O₁₁ are missing in the figure.
- XRD data presented only shows preliminary information and does not provide full evidence of a heterojunction formation between α -MoO₃ and η -Mo₄O₁₁. The data has to be revisited or even new experiments may have to be performed when the present data is unable to proof the claims.

Thin film preparation using sputtering: The manuscript does not provide any details of how the experiment was done. What were the targets used? How was the heterojunction designed during sputtering? Was such design controllable? The XRD pattern of the thin is rather confusing. In Figure 2(c) authors describe that the space group of α -MoO₃ is Pbnm, the α -MoO₃/ η -Mo₄O₁₁ heterojunction is constructed via P21-a/Pbnm [Figure 2(d)], the thin film of η -Mo₄O₁₁ is Pbnm (Supplementary Figure 1). Which are the correct space groups for each individual and interfaced materials? The structural evidence is very confusing and misleading.

Figure 3: Imaging of α -MoO₃/ η -Mo₄O₁₁ heterostructure: During imaging, particles tend to orient in various possible directions as presented in X-ray diffraction signals. Looking at Figure 3, the

heterojunction formation claim is superficial. Any material tends to have different orientation and this also depends on the crystallinity. Authors show purple spheres on the white spots of the HRTEM imaging. It seems that the possible Mo atoms is ambiguous. These white spots in ordered pattern are Mo-O polyhedra and these can be evident via supper positioning of the MoO₃ and/or η-Mo₄O₁₁ on the ordered pattern. Once again, this data is also not properly analyzed, incomplete or misleading.

Battery fabrication: How was the battery fabricated? What was done to avoid short circuit between MoO₃ and/or MoO₃/η-Mo₄O₁₁ and LIPON (electrolyte)? How was the electrolyte coated in between anode and cathode? How did LiPO₄ transformed into LiPON during sputtering? The details are not easily available in this manuscript.

The electrochemical performance of MoO₃/η-Mo₄O₁₁ is clearly stable and higher than the MoO₃. What are the reasons for this effect? Why did specific capacity spike at 700, 1100 and 1600 cycles? Authors are advised to correct all the physicochemical properties as commented earlier and reconnect the performance with the new analysis of their data.

Reviewer #2 (Remarks to the Author):

The article about lattice pinning in MoO₃ cathode and lithium ion diffusion study in Lithium ion battery. The concept of heterostructure and interface engineering by magnetron sputtering is interesting and useful. However, such concept is already available to many metal oxides in the literature. However, the current report on MoO₃/Mo₄O₁₁ is new in that sense. The use of MoO₃ microstructured in thin film lithium metal battery could be justified more with some more data. The microstructure and the interface formation is discussed nicely with experimental evidence. Only problem in the article was the use of many words like "ultrastable", coherent etc. which may be not completely justified in the experimental section.

The treatment of the data analysis and the discussion of the observed data are excellent. The presentation of the data and the figures are in higher level. I strongly recommend such article in current state.

Reviewer #3 (Remarks to the Author):

The manuscript details the groups study on MoO₃/Mo₄O₁₁ thin film materials.

1) I liked the concept of using a zero strain material as the interfacial species with the SSE, as noted early in the manuscript. Do you have a clear idea of how much volume expansion is too much? In some of these very restricted systems the tolerance for volume expansion (without polymers or liquids) may be very small.

2) you discuss around p6 the observation (SEM) that their are domains of Mo₄O₁₁ in MoO₃. If this is accurate - can it be described as an interfacial material where the new Mo₄O₁₁ is at the critical interface or a physical intergrowth - similar to the spinel-Li₂MnO₃ example you noted.

3) how coherent are the two phases? I did not get a good feeling from the discussion of the exact nature of this interface? It would seem that stress and strain issues would do more than extend the cycling but may also damage the interface - especially in lieu of comment 2

4) do you have any data to suggest how much energy can actually be stored - thin film batteries are usually very good due to their near perfect attachment to the current collector and thin dimensions. The small size can be deceptive if one is comparing bulk electrode data to thin film data (I know you are not - but most of the uses are bulk electrodes).

5) the 16% volume expansion of MoO₃ is in line with graphite. It may actually be workable but for the relatively low voltage - more of an issue with molybdenum oxide.

6) the "fixing" of the MoO₆ octahedra to a non-distorted octahedra is commonly seen on any

reduction of Mo(VI) center - im not sure its a significant result although it may help.

Point-by-point response to reviewers' comments

Reviewer #1 (Remarks to the Author):

This is an interesting manuscript demonstrating an attempt to redesign α -MoO₃/ η -Mo₄O₁₁ heterostructure as an alternative to avoid high lattice expansion during electrochemical cycling. Authors claim such heterojunction enhance Li⁺ mobility and storage with longer life cycle and/or specific capacity. Although this is a good work hypothesis, there are critical issues against the publication of this manuscript. While the journal is highly reputed, any misleading data either needs to be corrected, remeasured or clarified.

Response: We gratefully thank this reviewer for taking time to read through our manuscript and provide constructive criticisms. Accordingly, we have made substantial revisions to address each of these concerns.

Taking these facts into consideration, following critical issues are described below:

1. Authors have attempted to design a solid state interface with α -MoO₃ and isostructural η -Mo₄O₁₁ to reduce large lattice expansion of α -MoO₃ due to Li intercalation or deintercalation. They have focused only on zero-strain interfacial component. However, the solid state chemistry is not so trivial. The charge transfer (including Li intercalation) depends on the specific band positions of two interfacial materials having significant influence on the fermi energy for charge equilibration. Although zero strain material do have an impact on reducing lattice expansion, the mobility of majority carriers (either electrons or holes) across the space charge layers have permanent impact on the Li⁺ extraction or intercalation. These ions have to move following charge equilibration rule. Hence this manuscript has largely excluded the discussion on the solid-state interfacial chemistry during charging or discharging.

Response: We agree with the reviewer that solid state interfacial chemistry is important to charge transfer in the heterostructure. Therefore, we further analyzed and discussed

the influence of MoO₃/Mo₄O₁₁ interface on charge transfer during charge and discharge processes.

Compared with both α -MoO₃ and η -Mo₄O₁₁ electrodes, α -MoO₃/ η -Mo₄O₁₁ electrode possesses fast Li⁺ diffusion (Supplementary Fig. 12), which could be assigned to the solid state interfacial chemistry. Based on the ultraviolet photoelectron spectroscopy (UPS) and ultraviolet-visible absorption spectroscopy measurements (Supplementary Fig. 13a), we proposed the schematic energy-level diagrams of α -MoO₃, η -Mo₄O₁₁, and heterostructured α -MoO₃/ η -Mo₄O₁₁ in Supplementary Fig. 13b. Supplementary Fig. 13a displays the band structures of isolated α -MoO₃ and η -Mo₄O₁₁ before contact. The Fermi levels of both α -MoO₃ and η -Mo₄O₁₁ are all located near their Conduction Bands, suggesting their n-type semiconductor feature. As α -MoO₃ exhibits a smaller work function (4.28 eV) as compared to that of η -Mo₄O₁₁ (4.58 eV), electrons tend to transfer from α -MoO₃ to η -Mo₄O₁₁ across the interface. The electron flow leads to the accumulation of positive charge on the α -MoO₃ side and negative charge on the η -Mo₄O₁₁ side near the interface. Simultaneously, the energy levels of α -MoO₃ shift upward whereas those of η -Mo₄O₁₁ bend downward near the interface until their Fermi levels reach equilibrium [Xu, M., et al. ChemSusChem, 8, 1218–1225 (2015)]. Thus, a built-in electric field with a direction pointing from α -MoO₃ to η -Mo₄O₁₁ is formed at the α -MoO₃/ η -Mo₄O₁₁ heterojunction.

Given that η -Mo₄O₁₁ domains are randomly embedded within the α -MoO₃ matrix, Li⁺ ions need to be transferred from α -MoO₃ to η -Mo₄O₁₁ during the discharge process. During the discharge process, the as-formed built-in electric field can accelerate Li⁺ diffusion from α -MoO₃ to η -Mo₄O₁₁, thus improving the electrode kinetics for discharge process. Under the built-in electric field, Li⁺ could accumulate at η -Mo₄O₁₁ side to neutralize the negative charges, and the electric field around the hetero-interface may finally vanish after the charge balance [Fang, L., et al. Energy Storage Mater., 18, 107–113 (2019)]. Upon the charge process, Li⁺ ions are first extracted from α -MoO₃ owing to its direct contact with electrolyte. As α -MoO₃ possesses a more open structure for Li⁺ diffusion, the fast leaching of Li⁺ in α -MoO₃ will lead to a Li⁺ concentration gradient across the α -MoO₃/ η -Mo₄O₁₁ interface, thus generating a reversed built-in

electric field during the charge process. Under such an electric field, the transfer of Li^+ from $\eta\text{-Mo}_4\text{O}_{11}$ to $\alpha\text{-MoO}_3$ can be accelerated, improving the electrode kinetics for charge process [Zhang, C., et al. *Energy Storage Mater.*, 24, 208–219 (2020)]. Overall, the constructed $\alpha\text{-MoO}_3/\eta\text{-Mo}_4\text{O}_{11}$ hetero-interface is beneficial to improve Li^+ intercalation and deintercalation in this heterostructured electrode during both charge and discharge processes.

Supplementary Figure 12 Chemical diffusion coefficients of Li^+ (D_{Li^+}) of $\alpha\text{-MoO}_3/\eta\text{-Mo}_4\text{O}_{11}$, $\alpha\text{-MoO}_3$, and $\eta\text{-Mo}_4\text{O}_{11}$ electrodes during the lithiation process.

Supplementary Figure 13 a, Valence band spectra and corresponding band structure diagrams determined by ultraviolet photoelectron spectroscopy of $\eta\text{-Mo}_4\text{O}_{11}$, $\alpha\text{-MoO}_3/\eta\text{-Mo}_4\text{O}_{11}$, and $\alpha\text{-MoO}_3$ thin films. **b**, The schematic illustration of the built-in electric field between n-type $\alpha\text{-MoO}_3$ and n-type $\eta\text{-Mo}_4\text{O}_{11}$.

Supplementary Figures 12 and 13 have been added to the Supplementary Information, and related discussion has been added in the revised manuscript.

In the main Text (line 29 in Page 10 to line 6 in Page 12):

“Meanwhile, the chemical diffusion coefficient of lithium (D_{Li^+}) of α -MoO₃, η -Mo₄O₁₁, and α -MoO₃/ η -Mo₄O₁₁ electrodes were calculated (Supplementary Fig. 12) by conducting the galvanostatic intermittent titration (GITT) technique during the lithiation process. The D_{Li^+} values in α -MoO₃ electrode are substantially higher than those in η -Mo₄O₁₁ electrode owing to the large ionic diffusion channels. However, the irreversible phase transition results in the dramatic decrease of the diffusion coefficient of Li⁺ in α -MoO₃ electrode. Compared with both α -MoO₃ and η -Mo₄O₁₁ electrodes, α -MoO₃/ η -Mo₄O₁₁ electrode possesses fast and stable Li⁺ diffusion, which could be assigned to the solid state interfacial chemistry and improved structural stability.

Based on the ultraviolet photoelectron spectroscopy (UPS) and ultraviolet-visible absorption spectroscopy measurements (Supplementary Fig. 13a), the schematic energy-level diagrams of α -MoO₃, η -Mo₄O₁₁, and heterostructured α -MoO₃/ η -Mo₄O₁₁ were proposed as shown in Supplementary Fig. 13b. The Fermi levels of both α -MoO₃ and η -Mo₄O₁₁ are all located near their Conduction Bands, suggesting their n-type semiconductor feature. As α -MoO₃ exhibits smaller work function (4.28 eV) as compared to that of η -Mo₄O₁₁ (4.58 eV), electrons tend to transfer from α -MoO₃ to η -Mo₄O₁₁ across the interface. The electron flow leads to the accumulation of positive charge on the α -MoO₃ side and negative charge on the η -Mo₄O₁₁ side near the interface. Simultaneously, the energy levels of α -MoO₃ shift upward whereas those of η -Mo₄O₁₁ bend downward near the interface until their Fermi levels reach equilibrium.³² Thus, a built-in electric field with a direction pointing from α -MoO₃ to η -Mo₄O₁₁ is formed at the α -MoO₃/ η -Mo₄O₁₁ heterojunction. Given that η -Mo₄O₁₁ domains are randomly embedded within the α -MoO₃ matrix, Li⁺ ions need to be transferred from α -MoO₃ to η -Mo₄O₁₁ during the discharge process. During the discharge process, the as-formed built-in electric field can accelerate Li⁺ diffusion from α -MoO₃ to η -Mo₄O₁₁, thus improving the electrode kinetics for discharge process. Under the built-in electric field, Li⁺ could be accumulated at η -Mo₄O₁₁ side to neutralize the negative charges, and the electric field around the hetero-interface may finally vanish after the charge balance.³³ Upon the charge process, Li⁺ ions are first extracted from α -MoO₃ owing to its direct

contact with electrolyte. As α - MoO_3 possesses a more open structure for Li^+ diffusion, the fast leaching of Li^+ in α - MoO_3 will lead to a Li^+ concentration gradient across the α - MoO_3/η - Mo_4O_{11} interface, thus generating a reversed built-in electric field during the charge process. Under such an electric field, the transfer of Li^+ from η - Mo_4O_{11} to α - MoO_3 can be accelerated, improving the electrode kinetics for charge process.³⁴ Overall, the constructed α - MoO_3/η - Mo_4O_{11} hetero-interface is beneficial to improve Li^+ intercalation and deintercalation in this heterostructured electrode during both charge and discharge processes.”

2. Figure 2: X-ray diffraction analysis of α - MoO_3 . Authors are suggested to revisit the analysis to explain the following ambiguities.

The review of the data presented in Figure 2c with all the Pbnm space group of the orthorhombic system (α - MoO_3), none of the ICSD data matches with the x-ray diffraction signal intensity presented in the manuscript. The significant mismatch in the intensity of the signals corresponding at two theta 12.9, 23.3, 25.9,27.4 suggest the wrong assignment of the space group. When the XRD patterns are refined, what was the ICSD data (as a cif file used)? Why was JCPDS data (mainly used for the signal positions) given more focus? Authors are advised to give detail description of how they manage to refine this data with such intensity mismatch?

Response: We thank the reviewer for the insightful comments on the XRD analysis. The ICSD and JCPDS data are based on the X-ray diffraction data for standard powder samples without preferential orientation. The preferential orientation of crystallites in either powder or thin film samples could lead to variation in diffraction signal intensity in comparison with the standard powder sample [Abboudi, M., et al. Journal of Taibah University for Science, 12, 133–137 (2018); Uthanna, S., et al. Appl. Surf. Sci., 256, 3133–3137 (2010); Stefanov, B., et al. Coatings, 4, 587–601 (2014)]. Notably, this phenomenon is more significant in the thin film samples as they usually possess some preferential orientation [Hallot, M., et al. Energy Storage Mater., 15, 396–406 (2018); Lyandres, O., et al. Chem. Mater., 24, 3355–3362 (2012)]. For example, Lyandres *et al.*

compared the XRD patterns of RF sputtered anatase TiO₂ thin film and anatase TiO₂ powder, showing that the relative peak intensities of the thin film differ markedly from those of powder [Lyandres, O., et al. Chem. Mater., 24, 3355–3362 (2012)]. Uthanna *et al.* reported the influence of substrate temperature on the crystallographic texture of sputtered α -MoO₃ thin films [Uthanna, S., et al. Appl. Surf. Sci., 256, 3133–3137 (2010)]. As shown in Figure A1a, the α -MoO₃ thin film samples prepared at different substrate temperatures present different peak intensities. Similar intensity mismatch can also be observed in the XRD patterns of α -MoO₃ thin films reported by Martínez *et al.* (Figure A1b) [Martínez, H., et al. Mater. Charact., 75, 184–193 (2013)].

Figure A1 (a–b) X-ray diffraction patterns of α -MoO₃ films prepared at different substrate temperatures [Uthanna, S., et al. Appl. Surf. Sci., 256, 3133–3137 (2010); Martínez, H., Mater. Charact., 75, 184–193 (2013)].

The ICSD data used to refine the XRD patterns is α -MoO₃ (ICSD #36167). Considering the thin film texture effect, Rietveld refinement on thin film XRD patterns was carried out by using the harmonic texture model implemented into the Maud software [Zhou, X., et al. Nature, 579, 67–72 (2020); Kruszynska, M., et al. J. Am. Chem. Soc., 132, 15976–15986 (2010); Zhang, M.-H., Nat. Commun., 13, 3434 (2022)]. With the correction of the preferred orientation effect, we can get a good Rietveld refinement for XRD patterns of our thin film samples.

The corresponding description for refinement has been included in the Methods section of the main Text:

In the main Text (lines 20–23 in Page 22):

“The ICSD data used to refine the XRD patterns is α -MoO₃ (ICSD #36167). Considering the thin film texture effect, Rietveld refinement on thin film XRD patterns was carried out by using the harmonic texture model implemented into the Maud software.^{49–51}”

3. It was quite difficult to review this data while the intensity figures were removed. If authors want to publish their work in such a reputed journal, the data should be included in the figures or text as it comes from the instrument. Authors are advised to include the intensity data.

Response: We thank the reviewer for the constructive comment and suggestion, following which we have added the XRD data with the intensity to figures. Corresponding results have been incorporated into the Supplementary Information:

In the Supplementary Information:

Supplementary Figure 1 a–c, The XRD patterns of α -MoO₃, α -MoO₃/ η -Mo₄O₁₁, and η -Mo₄O₁₁ thin films with intensity data. The asterisk represents the diffraction peaks of the substrate.

4. While the signal at 12.9 two theta is characteristic signal for α -MoO₃ (as described in Figure 2c), why is it missing in the α -MoO₃/ η -Mo₄O₁₁ heterostructure [Figure 2(d)]?

Response: Thanks for the comments. The missing of the signal at 12.9 degree is due to the change of preferential orientation of the film. The preferential orientation of thin

film largely depends on the deposition conditions (e.g., substrate temperature and oxygen partial pressure) during the sputtering process [Windischmann, H., et al. Crit. Rev. in Solid State, 17, 547–596 (1992); Kim, J.-H., et al. J. Electroceram., 23, 169–174 (2009); Song, L., et al. ACS Appl. Energy Mater., 3, 2055–2062 (2020)]. In specific, the effects of the oxygen partial pressure on the preferential orientation of sputtered metal oxide thin films have been demonstrated in previous literature [Stefanov, B., et al. Coatings, 4, 587–601 (2014); Sérgio, S., et al. J. Alloy. Compd., 434, 701–703 (2007)]. The O₂/Ar flow ratio for the α -MoO₃/ η -Mo₄O₁₁ heterostructured thin film deposition was 13%, which was much lower than that of α -MoO₃ (25%) in Figure 2c. Therefore, the missing of (020) diffraction peak of α -MoO₃ in the XRD pattern of the α -MoO₃/ η -Mo₄O₁₁ heterostructure can be attributed to the variation in preferential orientation of the film.

5. If the pattern is refined using two different phases (α -MoO₃ and η -Mo₄O₁₁) what is the mass content of the two phases?

Response: Thanks for the comments. Based on the Rietveld refinement results, the mass contents of α -MoO₃ and η -Mo₄O₁₁ in the α -MoO₃/ η -Mo₄O₁₁ heterostructure are ~64 wt% and 36 wt%, respectively.

The corresponding description for mass content of the two phases has been included in the main Text:

In the main Text (lines 15–17 in Page 5):

“Via Rietveld refinement, Fig. 2d reveals that the heterostructure possesses approximately 64 wt% α -MoO₃ and 36 wt% η -Mo₄O₁₁, respectively, indicating α -MoO₃ is the major phase for the heterostructure.”

6. Many signals which are supposed to be present for the P21-a of η -Mo₄O₁₁ are missing in the figure.

Response: Thanks for the comments. As discussed in the *Response to Question 2 and*

4, the sputtered thin films exhibit stronger preferential orientation than powders, which could lead to the disappearance of some diffraction signals in XRD. As shown in the XRD pattern of η -Mo₄O₁₁, one dominant peak located at 22.7° along with three small peaks at 26.2°, 27.3 ° and 36.7° can be ascribed to the (211), (411), (1000) and (−1002) reflections of η -Mo₄O₁₁ (P21-a, JCPDS 13-0142 card), indicating a strong (211) preferred orientation of the η -Mo₄O₁₁ thin film. This phenomenon can also be found in reported Mo₄O₁₁ and other metal oxide thin films in literature [Guerrero, R. M., et al. *J. Alloy. Compd.*, 434, 701–703 (2007); Wang, Y., *Acta Mater.*, 76, 207–212 (2014); Turgut, E., et al. *Appl. Surf. Sci.*, 435, 88–885 (2018); Hassanien, A., et al. *Opt. Quant. Electron.*, 52, 194 (2020)].

7. XRD data presented only shows preliminary information and does not provide full evidence of a heterojunction formation between α -MoO₃ and η -Mo₄O₁₁. The data has to be revisited or even new experiments may have to be performed when the present data is unable to proof the claims.

Response: Thanks for the comments. We agree with the reviewer that XRD data alone cannot demonstrate the heterojunction formation between α -MoO₃ and η -Mo₄O₁₁. Therefore, in the present study, the heterojunction formation between α -MoO₃ and η -Mo₄O₁₁ was demonstrated by the high-resolution HAADF-STEM and HRTEM. As clearly shown in the HAADF-STEM image in Fig. 3d, the α -MoO₃ domain and the η -Mo₄O₁₁ domain in the film are connected via a coherent interface, thus creating the heterojunction between α -MoO₃ and η -Mo₄O₁₁. Moreover, the HRTEM image of the interface region of the α -MoO₃/ η -Mo₄O₁₁ sample along [100] zone axis of α -MoO₃ was also provided in Supplementary Figure 6, further demonstrating the heterojunction formation between α -MoO₃ and η -Mo₄O₁₁.

Fig. 3 d, HAADF-STEM image for the α -MoO₃/ η -Mo₄O₁₁ heterostructure. The blue square corresponds to the monoclinic Mo₄O₁₁ viewed along the [100] zone axis while the red square corresponds to the orthorhombic MoO₃ viewed along the [101] zone axis.

Supplementary Figure 6 a, HRTEM image for the α -MoO₃/ η -Mo₄O₁₁ sample. **b**, HRTEM image image for interface region of the α -MoO₃/ η -Mo₄O₁₁ sample along the [100] zone axis of α -MoO₃. FFT images for monoclinic η -Mo₄O₁₁ (collected from A area) of the α -MoO₃/ η -Mo₄O₁₁ sample along the [001] zone axis. FFT images for layered α -MoO₃ (collected from B area) of the α -MoO₃/ η -Mo₄O₁₁ sample along the [100] zone axis.

8. Thin film preparation using sputtering: The manuscript does not provide any details of how the experiment was done. What were the targets used?

Response: We thank the reviewer for the constructive question. The targets used for the preparation of Mo-based thin films are pure Mo targets (75 mm in diameter and 5 mm in thickness). The preparation details of Mo-based thin films using DC sputtering

have been provided in the Methods section of the main Text:

In the main Text (line 27 in Page 21 to line 12 in Page 22):

“The α -MoO₃, α -MoO₃/ η -Mo₄O₁₁, and η -Mo₄O₁₁ thin films were deposited on Pt/Ti/glass or Pt/Ti/stainless steel substrates using a pure Mo target (75 mm in diameter and 5 mm in thickness) through DC magnetron sputtering (SKY Technology Development Co., Ltd, China) under O₂ and Ar atmosphere. Before deposition, a background pressure of less than 1×10^{-5} Pa was first reached in the chamber. The distance between the target and substrate was kept at 10 cm and the substrate rotation is 20 rpm. The DC power was fixed at 60 W. The α -MoO₃ thin film was deposited at 573 K, in a working pressure of ~ 1 Pa introduced by mass flow controllers with 40.0 sccm Ar and 10.0 sccm O₂ (25% O₂/Ar ratio). α -MoO₃/ η -Mo₄O₁₁ thin film was deposited at 573 K, in a working pressure of ~ 1 Pa introduced by mass flow controllers with 40.0 sccm Ar and 5.0 sccm O₂ (13% O₂/Ar ratio). η -Mo₄O₁₁ thin film was deposited at 573 K, in a working pressure of ~ 1 Pa introduced by mass flow controllers with 40.0 sccm Ar and 3.0 sccm O₂ (8% O₂/Ar ratio). All the films were deposited for 1.5 h and then annealed in the chamber at 673 K for 1 h to obtain the final α -MoO₃, α -MoO₃/ η -Mo₄O₁₁, and η -Mo₄O₁₁ thin films.”

9. How was the heterojunction designed during sputtering? Was such design controllable?

Response: Thanks for the comments. During the sputtering process, the as-formed ionized Ar (Ar⁺) are driven towards the Mo target under the electric field. After Ar⁺ bombardment, the target Mo atoms are released to react with the O⁻ at 400–800 K and get deposited on the substrate in the form of Mo oxides [Xia, Q., et al. Small, 14, 1804149 (2018)]. Therefore, the chemical composition and Mo valence state of the film can be well controlled by tuning the oxygen partial pressure and substrate temperature during deposition. As shown in Fig. 2b, when the O₂/Ar flow ratio was larger than 20%, Mo atoms can be completely oxidized to form α -MoO₃. With the decrease of O₂/Ar flow ratio to less than 20%, low valence molybdenum oxides such as η -Mo₄O₁₁ were

gradually generated together with α -MoO₃, resulting in the formation of α -MoO₃/ η -Mo₄O₁₁ heterojunction. Such heterojunction design can be easily achieved by controlling the O₂/Ar flow ratio (10%~20%) and substrate temperature (573~673 K).

We have incorporated the above results and discussion into the main Text:

In the main Text (lines 25–28 in Page 4):

“During the sputtering process, the as-formed ionized Ar (Ar⁺) are driven towards the Mo target under the electric field. After Ar⁺ bombardment, the target Mo atoms are released to react with the O⁻ at 400–800 K and get deposited on the substrate in the form of Mo oxides.¹⁹”

(lines 1–8 in Page 5)

“The chemical composition and Mo valence state of the film can be well controlled by tuning the oxygen partial pressure and substrate temperature during deposition. As shown in Fig. 2b, when the O₂/Ar flow ratio was larger than 20%, Mo atoms can be completely oxidized to form α -MoO₃. With the decrease of O₂/Ar flow ratio to less than 20%, low valence molybdenum oxides such as η -Mo₄O₁₁ were gradually generated together with α -MoO₃, resulting in the formation of α -MoO₃/ η -Mo₄O₁₁ heterojunction. Such heterojunction design can be easily achieved by controlling the O₂/Ar flow ratio (10%~20%) and substrate temperature (573~673 K).”

10. The XRD pattern of the thin is rather confusing. In Figure 2(c) authors describe that the space group of α -MoO₃ is Pbnm, the α -MoO₃/ η -Mo₄O₁₁ heterojunction is constructed via P21-a/Pbnm [Figure 2(d)], the thin film of η -Mo₄O₁₁ is Pbnm (Supplementary Figure 1). Which are the correct space groups for each individual and interfaced materials? The structural evidence is very confusing and misleading.

Response: We appreciate the reviewer for the kind reminder and apologize for the confusion. The space groups of α -MoO₃, α -MoO₃/ η -Mo₄O₁₁, and η -Mo₄O₁₁ samples are Pbnm, P21-a/Pbnm, and P21-a, respectively. The error in Supplementary Fig. 2 has been corrected accordingly.

In the Supplementary Information:

Supplementary Figure 2 Rietveld-refined XRD pattern of the η -Mo₄O₁₁ thin film.

11. Figure 3: Imaging of α -MoO₃/ η -Mo₄O₁₁ heterostructure: During imaging, particles tend to orient in various possible directions as presented in X-ray diffraction signals. Looking at Figure 3, the heterojunction formation claim is superficial. Any material tends to have different orientation and this also depends on the crystallinity. Authors show purple spheres on the white spots of the HRTEM imaging. It seems that the possible Mo atoms is ambiguous. These white spots in ordered pattern are Mo-O polyhedra and these can be evident via superpositioning of the MoO₃ and/or η -Mo₄O₁₁ on the ordered pattern. Once again, this data is also not properly analyzed, incomplete or misleading.

Response: Thanks for the comments. The heterojunction formation between α -MoO₃ and η -Mo₄O₁₁ was determined by XRD, Raman, HRTEM, and HAADF-STEM. Fig. 3 presents the aberration-corrected HAADF-STEM images but not HRTEM images. Therefore, the white spots correspond to Mo atomic columns due to their large atomic number. However, oxygen atoms cannot be clearly observed due to their small atomic number with low contrast. We totally agree with the reviewer that simply putting the Mo symbols on the white spots cannot clearly present different structures of α -MoO₃ and η -Mo₄O₁₁. Therefore, the ideal structural models of α -MoO₃ and η -Mo₄O₁₁ are superimposed on the ordered pattern in Fig. 3, displaying good qualitative agreement. Moreover, corresponding FFT patterns for d1 and d2 regions in Fig. 3d further confirm the coexistence of two phases and the formation of heterojunction in the film. In the

Supplementary Information, the HRTEM imaging result was provided (Supplementary Figure 6), which agrees well with the HAADF-STEM imaging, illustrating the intergrowth of α -MoO₃ and η -Mo₄O₁₁ phases in the film with the coherent interface.

We have incorporated the above results and discussion into the main Text:

In the main Text (lines 4–6 in Page 6):

“Ideal structure models of α -MoO₃ and η -Mo₄O₁₁ superimposed on the HAADF-STEM images were plotted to better illustrate the intergrowth of α -MoO₃ and η -Mo₄O₁₁ phases and formation of heterojunction.”

Fig. 3| Interface for the α -MoO₃/ η -Mo₄O₁₁ heterostructure. **a**, HAADF-STEM image for layered α -MoO₃ along [101] zone axis. The purple spheres represent Mo atoms and the yellow spheres represent oxygen atoms. **b**, HAADF-STEM image for monoclinic η -Mo₄O₁₁ along [100] zone axis. **c**, Crystal structures of orthorhombic MoO₃ along [101] zone axis and monoclinic Mo₄O₁₁ along [100] zone axis. **d**, HAADF-STEM image for the α -MoO₃/ η -Mo₄O₁₁ heterostructure. The blue square corresponds to the monoclinic Mo₄O₁₁ viewed along the [100] zone axis while the red square corresponds to the orthorhombic MoO₃ viewed along the [101] zone axis. **e**, The O K-edge ELNES spectra of the α -MoO₃/ η -Mo₄O₁₁ heterostructure obtained along the

yellow arrow in the left STEM image.

12. Battery fabrication: How was the battery fabricated?

Response: We thank the reviewer for the good question and apologize for the missed details about battery fabrication in Methods. The fabrication details of as-prepared all-solid-state thin-film battery have been included in the Methods section of the main Text and Supplementary Information:

In the main Text (lines 13–26 in Page 23):

“The all-solid-state thin-film battery was fabricated through a layer-by-layer deposition procedure. Stainless steel foil with a Ti/Pt transition layer was used as the substrate, and the α -MoO₃/ η -Mo₄O₁₁ thin film cathode was first deposited by DC sputtering. The LiPON solid electrolyte film and lithium metal anode film were then sequentially deposited onto the thin film cathode (Supplementary Fig. 23). Typically, the LiPON electrolyte film with a thickness of 2.5 μ m was deposited onto the cathode by RF reactive magnetron sputtering (SKY Technology Development Co., Ltd, China) using a Li₃PO₄ target (75 mm in diameter and 5 mm in thickness) under N₂ atmosphere at 298 K for 15 h. Then, the lithium metal layer of approximately 2.5 μ m was deposited on the LiPON layer by thermal evaporation (SKY Technology Development Co., Ltd, China). After the deposition of 500 nm Cu current collector by RF reactive magnetron sputtering (SKY Technology Development Co., Ltd, China) using a Cu target, the all-solid-state thin-film microbattery was finally sealed with aluminum-plastic film.”

In the Supplementary Information:

Supplementary Figure 23 The fabricated process of all-solid-state thin-film microbattery.

13. What was done to avoid short circuit between MoO_3 and/or $\text{MoO}_3/\eta\text{-Mo}_4\text{O}_{11}$ and LiPON (electrolyte)? How was the electrolyte coated in between anode and cathode? How did LiPO_4 transformed into LiPON during sputtering? The details are not easily available in this manuscript.

Response: We appreciate the good questions from the reviewer. LiPON electrolyte film with high ionic conductivity of about $3 \times 10^{-6} \text{ S cm}^{-1}$ and low electric conductivity of about $1 \times 10^{-10} \text{ S cm}^{-1}$ has been successfully used as the solid electrolyte [Xia, Q., et al. *Small*, 14, 1804149 (2018); Haruta, M., et al. *Nano Lett.*, 15, 1498–1502 (2015)]. In the sandwich solid-state battery structure, the intermediate LiPON layer functions as both electrolyte and separator, preventing the short circuit between the cathode and anode. With concern to the preparation process, the area of LiPON is larger than those of the cathode and anode to avoid the short circuit at the edge, as shown in Supplementary Figure 23. Typically, the LiPON solid electrolyte film was deposited onto the $\alpha\text{-MoO}_3/\eta\text{-Mo}_4\text{O}_{11}$ cathode by RF reactive magnetron sputtering, following

which the subsequent lithium anode layer was deposited by thermal evaporation.

The LiPON was sputtered using a Li_3PO_4 target under N_2 atmosphere at 298 K. During the RF magnetron sputtering process, the introduced N_2 gas was split into nitrogen ions and electrons under the high electric field, resulting in a reactive nitrogen cation species. The ionized N species can react with the Li-P-O species to form LiPON [Xia, Q., et al. Adv. Mater., 2200538 (2022); Dai, W., et al. Mater. Futures., 1 032101(2022)].

We have incorporated the discussion into the main Text:

In the main Text (lines 17–24 in Page 23):

“Typically, the LiPON electrolyte film with a thickness of 2.5 μm was deposited onto the cathode by RF reactive magnetron sputtering (SKY Technology Development Co., Ltd, China) using a Li_3PO_4 target (75 mm in diameter and 5 mm in thickness) under N_2 atmosphere at 298 K for 15 h. It should be noted that the area of LiPON is larger than those of the cathode and anode to avoid the short circuit at the edge.”

(lines 20–24 in Page 18):

“The LiPON was sputtered using a Li_3PO_4 target under N_2 atmosphere at 298 K. During the RF magnetron sputtering process, the introduced N_2 gas was split into nitrogen ions and electrons under the high electric field, resulting in a reactive nitrogen cation species. The ionized N species can react with the Li-P-O species to form LiPON.^{47, 48}”

14. The electrochemical performance of $\text{MoO}_3/\eta\text{-Mo}_4\text{O}_{11}$ is clearly stable and higher than the MoO_3 . What are the reasons for this effect?

Response: Thanks for the insightful question. It has been demonstrated that the insertion of Li^+ ions into the layered $\alpha\text{-MoO}_3$ breaks the thermodynamically stable state and induces large lattice expansion ($\sim 16\%$), resulting in the irreversible phase transformation to acicular Li_xMoO_3 ($x\sim 0.25$) accompanied by the fast capacity fading. In contrast, the monoclinic $\eta\text{-Mo}_4\text{O}_{11}$ presents minimal and reversible lattice change during both lithiation and delithiation processes (Supplementary Fig. 17), suggesting that it could be a perfect structural stabilizer to constrain lattice variation of $\alpha\text{-MoO}_3$ in

heterostructure design. As expected, η -Mo₄O₁₁ domains in the intergrown α -MoO₃/ η -Mo₄O₁₁ heterostructure can serve as pin centers to effectively suppress the lattice expansion of α -MoO₃, eliminating the irreversible phase transition during the cycling process (Fig. 5). With such lattice pinning effect, the α -MoO₃/ η -Mo₄O₁₁ heterostructured cathode exhibited significantly improved cycle performance in comparison with the α -MoO₃ cathode.

The irreversible phase transition in α -MoO₃ leads to structural degradation, which reduces the available Li⁺ storage sites with compromised capacity. In addition, the structural degradation in α -MoO₃ also slows down its electrode kinetics. With superior structural stability and favorable built-in electric field for the heterostructure, the α -MoO₃/ η -Mo₄O₁₁ cathode allows efficient Li⁺ intercalation and deintercalation with improved electrode kinetics, thus resulting in enhanced Li⁺ storage capability as compared to α -MoO₃.

15. Why did specific capacity spike at 700, 1100 and 1600 cycles? Authors are advised to correct all the physicochemical properties as commented earlier and reconnect the performance with the new analysis of their data.

Response: Thanks for your question. The specific capacity spikes at 700, 1100, and 1600 cycles could be ascribed to the small temperature turbulence of the long-term testing environment. As the battery is highly sensitive to temperature, a small change of testing temperature could affect its specific capacity. Since the long-term cycling test takes long time, we cannot avoid small temperature turbulence in the test room.

We appreciate all the valuable comments of the reviewer and the manuscript has been carefully revised according to these comments.

Reviewer #2 (Remarks to the Author):

The article about lattice pinning in MoO₃ cathode and lithium ion diffusion study in Lithium ion battery. The concept of heterostructure and interface engineering by magnetron sputtering is interesting and useful. However, such concept is already

available to many metal oxides in the literature. However, the current report on MoO₃/Mo₄O₁₁ is new in that sense. The use of MoO₃ microstructured in thin film lithium metal battery could be justified more with some more data. The microstructure and the interface formation are discussed nicely with experimental evidence. Only problem in the article was the use of many words like “ultrastable”, coherent etc. which may be not completely justified in the experimental section.

The treatment of the data analysis and the discussion of the observed data are excellent. The presentation of the data and the figures are in higher level. I strongly recommend such article in current state.

Response: We appreciate the favorable recommendation and encouraging comments of this referee.

To justify the use of MoO₃ microstructured material in thin film lithium metal batteries, supplementary Fig. 28 compares some crucial properties of all-solid-state thin film batteries based on different cathodes, such as α -MoO₃/ η -Mo₄O₁₁, LiCoO₂ [Trask, J., et al. J. Power Sources, 350, 56 (2017)], and LiMn₂O₄ [Xia, Q., et al. Small, 14, 1804149 (2018)]. In contrast to traditional Li-contained cathode thin films (LiCoO₂ and LiMn₂O₄) that necessitate a high annealing temperature around 973 K, the α -MoO₃/ η -Mo₄O₁₁ as well as α -MoO₃ thin films can be well crystallized at a much lower annealing temperature of 673 K, making battery on-chip integration compatible without damaging the chip [Sun, S., et al. J. Materiomics, 5, 229–236 (2019)]. Moreover, the α -MoO₃/ η -Mo₄O₁₁ based solid-state microbattery can achieve superior electrochemical performance to the LiCoO₂ and LiMn₂O₄-based solid-state microbatteries, especially in specific capacity, max energy density, and cycle life. Importantly, for the α -MoO₃/ η -Mo₄O₁₁ and α -MoO₃ thin film deposition, a cheap Mo metal target can be directly used in sputtering, making the fabrication cost much lower than LiCoO₂ and LiMn₂O₄ thin films.

The justification of using MoO₃ microstructure in thin film microbattery has been added in the revised manuscript, and supplementary Fig.28 has been added in the Supplementary Information.

Supplementary Fig. 28 Radar plots illustrating crucial metrics to assess $\alpha\text{-MoO}_3/\eta\text{-Mo}_4\text{O}_{11}$, LiMn_2O_4 , and LiCoO_2 cathodes in all-solid-state thin film lithium microbatteries.

Furthermore, the MoO_3 thin film cathode has high compatibility with LiPON solid electrolyte and a sharp $\text{MoO}_3/\text{LiPON}$ interface can be achieved in the solid-state thin film microbattery without any interlayer formation [Sun, S., et al. *J. Materiomics*, 5, 229–236 (2019)]. For the LiCoO_2 and LiMn_2O_4 -based all-solid-state thin film lithium microbatteries, an unexpected disordered interfacial layer is formed between the cathode and LiPON electrolyte even before cycling [Xia, Q., et al. *Small*, 14, 1804149 (2018); Wang, Z., et al. *Nano Lett.*, 16, 3760–3767 (2016)]. The disordered interfacial layer for LiCoO_2 and LiMn_2O_4 -based microbatteries impedes Li^+ diffusion, inducing high impedance at the electrolyte–electrode interface. Such high impedance interfacial layer grows thicker during cycling, resulting in rapid capacity decay [Wang, Z., et al. *Nano Lett.*, 16, 3760–3767 (2016)]. Therefore, by using MoO_3 thin film cathode in solid-state microbatteries, a highly stable cathode/electrolyte interface can be achieved, guaranteeing good interface stability during cycling.

The inappropriate words like “ultrastable” have been corrected in the revised manuscript. We further discussed the feature of the coherent interface of $\alpha\text{-MoO}_3/\eta\text{-Mo}_4\text{O}_{11}$, and related discussion has been added in the revised manuscript.

In the main Text (lines 13–29 in Page 7)

“Typically, the α -MoO₃/ η -Mo₄O₁₁ heterostructure demonstrates a specific orientation relationship of (020) _{α} //(020) _{η} and [10–1] _{α} //[001] _{η} with a coherent interface between α -MoO₃ and η -Mo₄O₁₁ (Fig. 3d). Generally, the direct connection between α -MoO₃ (020) and η -Mo₄O₁₁ (020) planes will generate a large lattice mismatch (f) of ~18.8%, defined by $f=(d_{\alpha}-2d_{\eta})/d_{\alpha}$, which is energetically unfavorable to form a coherent interface.²⁴ HAADF-STEM image at the interfacial region shows that a gradual transition zone with a thickness of approximately 4–5 nm instead of a sharp interface separates the two phases, thereby enabling the gradual release of misfit strain. To ensure the atomic matching for the two phases, the intralayer distance increases as the interlayer distance decreases in α -MoO₃ to accommodate the lattice of η -Mo₄O₁₁ in the transition zone. The interfacial gradual transition zone is critical to reducing lattice mismatch between α -MoO₃ and η -Mo₄O₁₁, resulting in the construction of coherent interface between these two phases.”

Reviewer #3 (Remarks to the Author):

The manuscript details the groups study on MoO₃/Mo₄O₁₁ thin film materials.

1) I liked the concept of using a zero strain material as the interfacial species with the SSE, as noted early in the manuscript. Do you have a clear idea of how much volume expansion is too much? In some of these very restricted systems the tolerance for volume expansion (without polymers or liquids) may be very small.

Response: Thanks for the insightful comments. Most of the current commercial cathodes undergo notable volume changes during the lithiation/delithiation processes, resulting in large strain and stress at the cathode/solid electrolyte interface in solid-state batteries. Depending on the cutoff voltage, layered LiNi_{0.8}Co_{0.1}Mn_{0.1}O₂ (NCM811), LiFePO₄, and spinel LiMn₂O₄ could undergo volume changes of approximately 6%, 7%, and 9%, respectively [Zhao, X., et al. Joule, 6, 1654–1671 (2022)]. In contrast, several layered cathodes such as LiNi_{0.3}Co_{0.6}Mn_{0.1}O₂ (NCM361) and LiNi_{0.2}Co_{0.7}Mn_{0.1}O₂ (NCM271) experience relatively smaller volume changes varying

from 0.2%~1% during charge up to 4.5 V vs. Li⁺/Li [Strauss, F., et al. ACS Mater. Lett., 2, 84–88 (2019)]. However, both the experimental results and theoretical calculation demonstrated that even very small volume changes in electrode materials can result in significant pressure/stress in solid-state batteries [Koerver, R., et al. Energy Environ. Sci., 11, 2142–2158 (2018); Zhang, W., et al. J. Mater. Chem. A, 5, 9929–9936 (2017); Yadav, N. G., et al. J. Mater. Chem. A, 10, 17142–17155 (2022)]. Janek and his coworkers estimated the hydrostatic pressure brought by volume expansion of cathode materials in LiCoO₂ based all-solid-state cells, in which the zero-strain Li₄Ti₅O₁₂ anode was used [Koerver, R., et al. Energy Environ. Sci., 11, 2142–2158 (2018); Zhang, W., et al. J. Mater. Chem. A, 5, 9929–9936 (2017)]. High pressure of approximately 0.6 MPa was calculated based on the volume change of +1.4% for LiCoO₂ upon delithiation, demonstrating the huge effect of very small volume change of the cathode on the cell pressure. Nevertheless, whether the cathode/solid electrolyte interface can withstand the high pressure or stress induced by electrode volume change also depends on the cell configuration, size, and interfacial chemistry. Therefore, it is hard to confirm how much volume expansion is too much for various solid-state battery systems. Nevertheless, the “zero-strain” materials could be ideal cathode candidates for all-solid-state lithium batteries.

2) you discuss around p6 the observation (SEM) that there are domains of Mo₄O₁₁ in MoO₃. If this is accurate-can it be described as an interfacial material where the new Mo₄O₁₁ is at the critical interface or a physical intergrowth-similar to the spinel-Li₂MnO₃ example you noted.

Response: We thank the reviewer for the insightful comments. The α-MoO₃/η-Mo₄O₁₁ thin film is composed of nanograins (approximately 30~100 nm) with irregular shapes according to the low-magnification HAADF-STEM image in Supplementary Fig. 5. Nano-domains of η-Mo₄O₁₁ along with intergrown hetero-interface have been demonstrated to be distributed inside α-MoO₃ nanograins (Fig. 3d–e). Just as mentioned by the reviewer, it is a physical intergrowth of α-MoO₃ and η-Mo₄O₁₁ in the film. The

intergrowth of the two phases occurs with a certain orientation relationship of $(020)_\alpha // (020)_\eta$ and $[10-1]_\alpha // [001]_\eta$ for α -MoO₃ and η -Mo₄O₁₁, thus creating the coherent interface.

Supplementary Figure 5 a,b, HAADF-STEM images for the α -MoO₃/ η -Mo₄O₁₁ sample at the low magnification.

Fig. 3d, HAADF-STEM image for the α -MoO₃/ η -Mo₄O₁₁ heterostructure. The blue square corresponds to the monoclinic Mo₄O₁₁ viewed along the [100] zone axis while the red square corresponds to the orthorhombic MoO₃ viewed along the [101] zone axis. **e**, The O *K*-edge ELNES spectra of the α -MoO₃/ η -Mo₄O₁₁ heterostructure obtained along the yellow arrow in the left STEM image.

3) how coherent are the two phases? I did not get a good feeling from the discussion of the exact nature of this interface? It would seem that stress and strain issues would do more than extend the cycling but may also damage the interface-especially in lieu of comment 2

Response: Thanks for your good questions. Typically, the α -MoO₃/ η -Mo₄O₁₁ heterostructure demonstrates a specific orientation relationship of $(020)_\alpha // (020)_\eta$ and $[10\bar{1}]_\alpha // [001]_\eta$ with a coherent interface between α -MoO₃ and η -Mo₄O₁₁ (Fig. 3d). Generally, the direct joining between α -MoO₃ (020) and η -Mo₄O₁₁ (020) planes (or a sharp interface) will generate a large lattice mismatch (f) of $\sim 18.8\%$, defined by $f = (d_\alpha - 2d_\eta) / d_\alpha$, which is energetically unfavorable to form a coherent interface [Sun, X., et al. Nature, 607, 708–713 (2022)]. However, the HAADF-STEM image at the interfacial region demonstrates that a gradual transition zone with a thickness of approximately 4–5 nm instead of a sharp interface separates the two phases, thereby enabling the gradual release of misfit strain. In contrast to sharp interface, the gradual transition interface enables the gradual release of misfit strain between the two lattices, leading to a continuous lattice mismatch of less than 5% at the interfacial region. Therefore, the gradual transition interface is critical to reducing lattice mismatch between α -MoO₃ and η -Mo₄O₁₁, resulting in the construction of coherent interface between these two phases. Simultaneously, the large stress and strain issues associated with sharp interface can be effectively mitigated by the gradual transition interface design, further improving the structural stability of the heterostructure.

Accordingly, further discussion about the exact nature of the hetero-interface in α -MoO₃/ η -Mo₄O₁₁ has been added in the revised manuscript. Details can be seen in the main Text:

In the main Text (lines 7–10 in Page 7):

“An interfacial region in the α -MoO₃/ η -Mo₄O₁₁ heterostructure is shown in Fig. 3d, in which η -Mo₄O₁₁ nanodomains of approximately 3–5 nm are found to be embedded in the layered α -MoO₃ matrix (Fig. 3d and Supplementary Fig. 6).”

(lines 13–29 in Page 7):

“Typically, the α -MoO₃/ η -Mo₄O₁₁ heterostructure demonstrates a specific orientation relationship of $(020)_\alpha // (020)_\eta$ and $[10\bar{1}]_\alpha // [001]_\eta$ with a coherent interface between α -MoO₃ and η -Mo₄O₁₁ (Fig. 3d). Generally, the direct joining between α -MoO₃ (020) and η -Mo₄O₁₁ (020) planes (or a sharp interface) will generate a large lattice mismatch (f)

of $\sim 18.8\%$, defined by $f=(d_\alpha-2d_\eta)/d_\alpha$, which is energetically unfavorable to form a coherent interface.²⁴ However, the HAADF-STEM image at the interfacial region demonstrates that a gradual transition zone with a thickness of approximately 4~5 nm instead of a sharp interface separates the two phases, thereby enabling the gradual release of misfit strain. In contrast to sharp interface, the gradual transition interface enables the gradual release of misfit strain between the two lattices, leading to a continuous lattice mismatch of less than 5% at the interfacial region. Therefore, the gradual transition interface is critical to reducing lattice mismatch between α -MoO₃ and η -Mo₄O₁₁, resulting in the construction of coherent interface between these two phases. Simultaneously, the large stress and strain issues associated with sharp interface can be effectively mitigated by the gradual transition interface design, further improving the structural stability of the heterostructure.”

Fig. 3d, HAADF-STEM image for the α -MoO₃/ η -Mo₄O₁₁ heterostructure. The blue square corresponds to the monoclinic Mo₄O₁₁ viewed along the [100] zone axis while the red square corresponds to the orthorhombic MoO₃ viewed along the [101] zone axis.

4) do you have any data to suggest how much energy can actually be stored-thin film batteries are usually very good due to their near perfect attachment to the current collector and thin dimensions. The small size can be deceptive if one is comparing bulk electrode data to thin film data (I know you are not-but most of the uses are bulk electrodes).

Response: Thanks for the valuable comments. Based on the charge and discharge

curves at a small current density of 0.05 A g^{-1} , the energy that can be stored in the α - $\text{MoO}_3/\eta\text{-Mo}_4\text{O}_{11}$ -based thin film battery with current dimension was calculated to be 0.48 mWh or $\sim 0.16 \text{ mWh cm}^{-2}$. The calculated specific energy per area of the thin film microbattery has been added in the revised manuscript.

In the main Text (lines 23–25 in Page 20):

“Based on the Galvanostatic charge/discharge profile in Supplementary Fig. 25, the α - $\text{MoO}_3/\eta\text{-Mo}_4\text{O}_{11}$ -based thin film microbattery delivers a specific energy density of $\sim 0.48 \text{ mWh}$ with an areal energy density of 0.16 mWh cm^{-2} .”

5) the 16% volume expansion of MoO_3 is in line with graphite. It may actually be workable but for the relatively low voltage-more of an issue with molybdenum oxide.

Response: Thanks for the valuable comments. We agree with the reviewer that the volume expansion of MoO_3 is similar to graphite and the low operating voltage is not beneficial to the cathode. Despite large volume expansion, it should be noted that the layered structure of graphite is highly stable to accommodate the reversible Li^+ ions insertion and extraction [Son, D.-K., et al. Carbon, 175, 187–201 (2021)]. In strong contrast, the large volume expansion of α - MoO_3 during initial Li^+ insertion causes irreversible conversion to acicular Li_xMoO_3 ($x \sim 0.25$). The irreversible phase transformation brings about large non-homogeneity of the strain inside MoO_3 , resulting in the accumulation of microcracks and fractures in the MoO_3 particles during the cycles, as demonstrated in supplementary Fig. 15. Therefore, the large volume expansion in α - MoO_3 is highly unfavorable, which leads to severe structural degradation and fast capacity fading during cycling.

Although the operating voltage for α - MoO_3 is relatively lower than those of traditional cathodes like LiCoO_2 and LiMn_2O_4 , the specific capacity of α - MoO_3 is much larger. Based on the galvanostatic charge/discharge profile in Supplementary Fig. 25, the α - $\text{MoO}_3/\eta\text{-Mo}_4\text{O}_{11}$ cathode in solid-state microbattery delivers a high specific energy density of $\sim 633 \text{ Wh kg}^{-1}$ (or 1143 Wh L^{-1}), which is larger than $\sim 600 \text{ Wh kg}^{-1}$ of Li_xMnO_2 [Xia, Q., et al. Adv. Mater., 33, 2003524 (2021)], $\sim 500 \text{ Wh kg}^{-1}$ of LiCoO_2

[Trask, J., et al. *J. Power Sources*, 350, 56 (2017)], and $\sim 400 \text{ Wh kg}^{-1}$ of LiMn_2O_4 [Xia, Q., et al. *Small*, 14, 1804149 (2018)] in solid-state thin film microbatteries.

Supplementary Fig. 15 have been added into the Supplementary Information:

In the Supplementary Information:

Supplementary Figure 15 a,b, TEM images of $\alpha\text{-MoO}_3/\eta\text{-Mo}_4\text{O}_{11}$ and $\alpha\text{-MoO}_3$ thin films collected at 5th cycles.

6) the "fixing" of the MoO_6 octahedra to a non-distorted octahedra is commonly seen on any reduction of Mo(VI) center – I'm not sure it's a significant result although it may help.

Response: Thanks for the valuable comments. The off-center displacement of d^0 Mo in MoO_6 octahedra of $\alpha\text{-MoO}_3$ (Mo^{6+}) is thermodynamically stable through the interaction between the empty d-orbitals (d^0) of Mo cations and filled p-orbitals of oxygen ions [Halasyamani, P. S., et al. *Chem. Mater.*, 16, 3586–3592 (2004); Meng, C.-Y., et al. *Dalton T.*, 46, 12320–12327 (2017); Inaguma, Y., et al. *J. Am. Chem. Soc.*, 136, 2748–2756 (2014)]. We agree with the reviewer that the "fixing" of the MoO_6 octahedra to a non-distorted octahedra is commonly seen on any reduction of Mo (VI) center. However, what we found based on the DFT calculations is that MoO_6 octahedra retain their distorted state without Mo migration back to the octahedral center when the lattice of $\alpha\text{-MoO}_3$ is pinned during lithiation or reduction of Mo (VI) center. As demonstrated in DFT calculations (Fig. 6a–b), the lithiation of $\alpha\text{-MoO}_3$ (Mo^{6+}) without lattice pinning, *i.e.*, the reduction of Mo(VI) center with the migration of Mo back to the octahedral

center, induces lattice expansion along b direction., driving the irreversible phase transition to form the unstable Li_xMoO_3 phase. In sharp contrast, by suppressing the lattice expansion along the b direction via the lattice pinning strategy, the "fixing" of the MoO_6 octahedra to the distorted state during the reduction of Mo(VI) center suppresses the irreversible phase transition of $\alpha\text{-MoO}_3$, enabling the $\alpha\text{-MoO}_3/\eta\text{-Mo}_4\text{O}_{11}$ heterostructure to present outstanding long-term cycling stability (Fig. 4f and Fig. 7f).

Reviewer comments, second round -

Reviewer #1 (Remarks to the Author):

This manuscript describes the heterojunction between α -MoO₃ and η Mo₄O₁₁ allows fast Li⁺ transport with very low sample degradation. The manuscript is a revised version of the paper and authors have attempted to answer most of the questions raised by the reviewers. However, Authors still need to look into the following point very carefully before this manuscript can be published.

1. This reviewer has no idea how the samples have been Rietveld refined to obtain α -MoO₃ and η Mo₄O₁₁ in the ratio of 64/36. The refinement process has not been described in the methods. What was the ICSD data base used? JCPDS No. 13-0142 and JCPDS No. 05-0508 are the crystallographic files where authors can observe if their patterns can be matched. They cannot however, be used in the Rietveld. Please include these results in manuscript discussion. What was the standard deviation obtained during refinement?

2. How were α -MoO₃ and η Mo₄O₁₁ targets acquired for magnetron sputtering? Are these materials industrially available? How did the material change (particle size, crystallinity etc.) after sputtering? These information are very important for reproducibility of the experiments. Authors are suggested to add this in their revised manuscript.

Reviewer #3 (Remarks to the Author):

Thank you for the comments.

1) I'm not convinced that such an advance in electrode materials design is as stable to cycling at the critical interfaces as needed.

2) the XRD pattern issues (at least to me) are in line with an incomplete characterization of the materials as prepared and evaluated.

Point-by-point response to reviewers' comments

Reviewer #1 (Remarks to the Author):

This manuscript describes the heterojunction between α -MoO₃ and η -Mo₄O₁₁ allows fast Li⁺ transport with very low sample degradation. The manuscript is a revised version of the paper and authors have attempted to answer most of the questions raised by the reviewers. However, Authors still need to look into the following point very carefully before this manuscript can be published.

Response: We appreciate the reviewer for his acknowledgment of our efforts put into the revision. Herein, we provide our point-to-point responses to his further comments.

1. This reviewer has no idea how the samples have been Rietveld refined to obtain α -MoO₃ and η -Mo₄O₁₁ in the ratio of 64/36. The refinement process has not been described in the methods. What was the ICSD data base used? JCPDS No. 13-0142 and JCPDS No. 05-0508 are the crystallographic files where authors can observe if their patterns can be matched. They cannot however, be used in the Rietveld. Please include these results in manuscript discussion. What was the standard deviation obtained during refinement?

Response: Thanks for the comments. One of the salient features of Rietveld refinement is the determination of phase fractions from the XRD data of a multiphase material. For the Rietveld refinement process with materials analysis using diffraction (MAUD) software, data is first refined according to the background and instrument parameters. Then, we used the structural model of α -MoO₃ ($a = 3.95 \text{ \AA}$, $b = 13.83 \text{ \AA}$, $c = 3.69 \text{ \AA}$, $Z = 4$, space group $Pbnm$, ICSD no. 36167) as the starting model in the refinement. Moreover, "excess" peaks that cannot be indexed to α -MoO₃ (ICSD no. 36167) were further refined as the second phase using the structural model of η -Mo₄O₁₁ ($a = 24.54 \text{ \AA}$, $b = 5.44 \text{ \AA}$, $c = 6.70 \text{ \AA}$, $Z = 4$, space group $P2_1/a$, ICSD no.

15814), generating a low reliability factor R_{wp} . Specifically, the quantitative value (64/36) of weight fraction between α -MoO₃ and η -Mo₄O₁₁ is extracted from the MAUD based on the following equation [Sahu, P. et al. J. Alloy Compd., 346, 158–169 (2002); Hill, R. J. et al. J. Appl. Cryst., 20, 467–474 (1987)]:

$$W_i = \frac{S_i(ZMV)_i}{\sum_{j=1}^N S_j(ZMV)_j}$$

where V , M , Z , and S are the unit-cell volume of phase i , the molecular weight, the number of molecules per unit cell, and the scale factor in the α -MoO₃/ η -Mo₄O₁₁ sample, respectively. The refinement process and how to obtain weight fractions have been provided in the Methods section of the main Text:

In the main Text (line 21 in Page 23 to line 6 in Page 24):

For the Rietveld refinement process with MAUD, data is first refined according to the background and instrument parameters. Then, we used the structural model of α -MoO₃ ($a = 3.95 \text{ \AA}$, $b = 13.83 \text{ \AA}$, $c = 3.69 \text{ \AA}$, $Z = 4$, space group $Pbnm$, ICSD no. 36167) as the starting model in the refinement. Moreover, “excess” peaks that cannot be indexed to α -MoO₃ (ICSD no. 36167) were further refined using the structural model of η -Mo₄O₁₁ ($a = 24.54 \text{ \AA}$, $b = 5.44 \text{ \AA}$, $c = 6.70 \text{ \AA}$, $Z = 4$, space group $P2_1/a$, ICSD no. 15814), generating a low reliability factor R_{wp} . Specifically, the quantitative value (64/36) of weight fraction between α -MoO₃ and η -Mo₄O₁₁ is extracted from the MAUD based on the following equation.^{50, 51}

$$W_i = \frac{S_i(ZMV)_i}{\sum_{j=1}^N S_j(ZMV)_j}$$

where V , M , Z , and S are the unit-cell volume of phase i , the molecular weight, the number of molecules per unit cell, and scale factor in the α -MoO₃/ η -Mo₄O₁₁ sample, respectively. Considering the thin film texture effect, Rietveld refinement on thin film XRD patterns was carried out by using the harmonic texture model implemented into the MAUD software.⁵²⁻⁵⁴

The crystal structure models used for the Rietveld refinement are available in the

ICSD (Inorganic Crystal Structure Database) under the following reference numbers (Supplementary Table 1): ICSD #36167 (space group *Pbnm*) for α -MoO₃ and ICSD #15814 (space group *P2₁/a*) for η -Mo₄O₁₁ [Tomaszewski, P. E. Phase Transit. 38, 127-220 (1992); Nugrahaningtyas, K. D. et al. Open Chem. 17, 1061-1070 (2019)]. The standard deviation obtained during refinement for α -MoO₃, α -MoO₃/ η -Mo₄O₁₁, and η -Mo₄O₁₁ has been included in Supplementary Table 2 [Takada, K. et al. Nature 422, 53-55 (2003); Gibot, P. et al. Nat. Mater. 7, 741-747 (2008)]. Based on the reviewer's comment, we revised the manuscript with the above details:

Supplementary Table 1 ICSD structures used for Rietveld refinement of α -MoO₃, α -MoO₃/ η -Mo₄O₁₁, and η -Mo₄O₁₁. (This table is added in the Supplementary table 1)

Phase	Space group	ICSD code	References
α -MoO ₃	Pbnm	36167	1
η -Mo ₄ O ₁₁	P2₁/a	15814	2

In the main Text (lines 18–20 in Page 5):

The standard deviation obtained during refinement for α -MoO₃, α -MoO₃/ η -Mo₄O₁₁, and η -Mo₄O₁₁ is shown in Supplementary Table 2. The value in bracket shows the standard deviation.²²

Supplementary Table 2 The refined crystallographic parameters of the cathode materials by the XRD patterns. (This table is added in the Supplementary table 2)

Sample	a(Å)	b(Å)	c(Å)	V(Å ³)	R _w p(%)	R _{exp} (%)
α -MoO ₃	3.9538(4)	13.8944(8)	3.6909(6)	202.7(7)	7.0	8.4
η -Mo ₄ O ₁₁	24.2538(1)	5.6786(5)	6.7210(9)	923.1(1)	9.6	6.9
α -MoO ₃ / η -Mo ₄ O ₁₁ (η)	24.3090(0)	5.6520(1)	6.7867(9)	928.0(8)	8.3	7.0
α -MoO ₃ / η -Mo ₄ O ₁₁ (α)	3.9630(3)	13.9243(7)	3.6942(0)	205.0(4)	8.3	7.0

2. How were α -MoO₃ and η -Mo₄O₁₁ targets acquired for magnetron sputtering? Are these materials industrially available? How did the material change (particle size, crystallinity etc.) after sputtering? These information are very important for reproducibility of the experiments. Authors are suggested to add this in their revised manuscript.

Response: Thanks for the valuable comments. Both α -MoO₃ and η -Mo₄O₁₁ thin films were deposited by using a pure Mo target (75 mm in diameter and 5 mm in thickness) through DC magnetron sputtering. The Mo target was industrially available material, which was purchased from Wuxi Kaistar Electro-optic Material Co., LTD. During the sputtering process, the as-formed ionized Ar (Ar⁺) are driven towards the Mo target under the electric field. After Ar⁺ bombardment, the Mo atoms are released from the target to react with the O⁻ at ~400–800 K and get deposited on the substrate in the form of Mo oxides. During the thin film deposition, the increment of oxygen partial pressure not only induces the phase transition from η -Mo₄O₁₁ to α -MoO₃ but also increases the surface roughness of the film. Supplementary Figure 5 shows the SEM images of Mo oxides obtained at different O₂/Ar flow ratios. At a low O₂/Ar flow ratio, the film exhibits a highly dense and smooth surface morphology, whereas at a

high O₂/Ar flow ratio, the film is composed of nanoflakes, resulting in a rough surface morphology. Therefore, increasing the O₂/Ar flow ratio can enlarge particle size, change phase composition, and enhance the valence state of Mo ions of the films.

The corresponding description has been included in the main Text:

In the main Text (lines 27–28 in Page 22):

The Mo target is industrially available material, which was purchased from Wuxi Kaistar Electro-optic Material Co., LTD.

In the main Text (lines 26 in Page 5 to lines 2 in Page 6):

As shown in Supplementary Fig. 5a–c, the increment of oxygen partial pressure not only induces the phase transition from η -Mo₄O₁₁ to α -MoO₃ but also increases the surface roughness of the film. At a low O₂/Ar flow ratio, the film exhibits a highly dense and smooth surface morphology, whereas at a high O₂/Ar flow ratio, the film is composed of nanoflakes, resulting in a rough surface morphology. Therefore, increasing the O₂/Ar flow ratio can enlarge particle size, change phase composition, and enhance the valence state of Mo ions of the films.

Supplementary Figure 5 a–c, FESEM images of η -Mo₄O₁₁, α -MoO₃/ η -Mo₄O₁₁, and α -MoO₃ thin films.

Reviewer #3 (Remarks to the Author):

Thank you for the comments.

1) I am not convinced that such an advance in electrode materials design is as stable

to cycling at the critical interfaces as needed.

Response: Thanks for the comments and we understand the reviewer's concern. To further demonstrate the high stability of the heterostructure and interface, we have characterized the α -MoO₃/ η -Mo₄O₁₁ thin film sample after 100 cycles. Prior to the cycling test, the HRTEM image of the interface region of the α -MoO₃/ η -Mo₄O₁₁ sample along [100] zone axis of α -MoO₃ was shown in Supplementary Figure 7, demonstrating the formation of a heterojunction between α -MoO₃ and η -Mo₄O₁₁. After 100 cycles at a specific current of 0.5 A g⁻¹, the HAADF-STEM imaging demonstrates the α -MoO₃/ η -Mo₄O₁₁ interface is well retained, as displayed in Supplementary Fig.21, indicating the high stability of the interface and heterostructure. Specifically, the Mo atomic arrangements in the red square agree well with the orthorhombic α -MoO₃ viewed along [100] zone axis. Meanwhile, the Mo atomic arrangements in the yellow square agree well with η -Mo₄O₁₁ viewed along [001] zone axis. This suggests that the coherent interface between α -MoO₃ and η -Mo₄O₁₁ is highly stable during the cycling process. In addition, to further verify the interfacial stability of the α -MoO₃/ η -Mo₄O₁₁ sample during cycling, we compared the differential capacity (dQ/dV) curves of the α -MoO₃/ η -Mo₄O₁₁ electrode at the 1st and 1000th cycles. As depicted in Supplementary Fig. 15, the redox peaks are well retained after 1000 cycles and the dQ/dV plots of α -MoO₃/ η -Mo₄O₁₁ electrode at the 1st and 1000th cycles are nearly overlapped with minimal capacity loss, further demonstrating the high stability of the heterostructure with coherent interface. In sharp contrast, the redox peaks in dQ/dV plot of α -MoO₃ are remarkably reduced and new peaks emerge even after 300 cycles, suggesting severe structural change.

The corresponding discussion has been included in the main Text:

In the main Text (line 18 in Page 16 to line 4 in Page 17):

The atomic structure of the cycled α -MoO₃/ η -Mo₄O₁₁ sample, which underwent 100 cycles at a specific current of 0.5 A g⁻¹, was further analyzed by HAADF-STEM. As seen in Supplementary Fig.21, both α -MoO₃ and η -Mo₄O₁₁ phases connected by the coherent interface can be clearly observed. The Mo atomic arrangements in the red

square agree well with the orthorhombic α -MoO₃ viewed along [100] zone axis, while the Mo atomic arrangements in the yellow square agree well with η -Mo₄O₁₁ viewed along [001] zone axis, demonstrating that the coherent interface between α -MoO₃ and η -Mo₄O₁₁ is highly stable during the cycling process.

In the main Text (lines 5–10 in Page 14):

In addition, as depicted in Supplementary Fig. 15, the redox peaks are well retained after 1000 cycles and the dQ/dV plots of α -MoO₃/ η -Mo₄O₁₁ electrode at the 1st and 1000th cycles are nearly overlapped with minimal capacity loss, further demonstrating the high stability of the heterostructure with coherent interface. In sharp contrast, the redox peaks in dQ/dV plot of α -MoO₃ are remarkably reduced and new peaks emerge even after 300 cycles, suggesting severe structural change.

Supplementary Figure 7 a, HRTEM image for the α -MoO₃/ η -Mo₄O₁₁ sample. **b**, HRTEM image for interface region of the α -MoO₃/ η -Mo₄O₁₁ sample along the [100] zone axis of α -MoO₃. FFT images for monoclinic η -Mo₄O₁₁ (collected from A area) of the α -MoO₃/ η -Mo₄O₁₁ sample along the [001] zone axis.

Supplementary Figure 21 a, HAADF-STEM image for the α -MoO₃/ η -Mo₄O₁₁ electrode collected after 100 cycles at a specific current of 0.5 A g⁻¹. **b**, Atomic arrangement in the red square represents orthorhombic α -MoO₃ along [100] zone axis. **c**, Atomic arrangement in the yellow square represents monoclinic η -Mo₄O₁₁ along [001] zone axis. The purple spheres represent Mo atoms and the yellow spheres represent oxygen atoms. (This figure is added in the Supplementary Figure 21)

Supplementary Figure 15 a,c Galvanostatic charge/discharge profiles of the $\alpha\text{-MoO}_3$ and $\alpha\text{-MoO}_3/\eta\text{-Mo}_4\text{O}_{11}$ cathodes collected at different cycle numbers. **b,d**, Calculated dQ/dV profiles of the $\alpha\text{-MoO}_3$ and $\alpha\text{-MoO}_3/\eta\text{-Mo}_4\text{O}_{11}$ electrodes after different cycle numbers at a specific current of 0.5 A g^{-1} . (This figure is added in the Supplementary Figure 15)

2) the XRD pattern issues (at least to me) are in line with an incomplete characterization of the materials as prepared and evaluated.

Response: Thanks for the comments. The XRD pattern issues that raised by the Reviewer #1 in the first revision include the intensity mismatch in reflections, the absence of some characteristic reflections related to $Pbnm$ of $\alpha\text{-MoO}_3$ and/or $P2_1/a$ of $\eta\text{-Mo}_4\text{O}_{11}$ in heterostructures, missed intensity figures, lack of information on the mass content of the two phases, additional data that confirms the formation of

heterojunction, and corresponding details about Rietveld refinement. These issues have been well settled in the 1st revision by providing sufficient details about Rietveld refinement, adding intensity figures, determining the mass content of the two phases through MAUD software, and re-analyzing the high-resolution HAADF-STEM and HRTEM.

Concerning the issues of intensity mismatch in reflections and the absence of certain reflections, we attributed these phenomena to the preferential orientation of the thin-film materials, based on our previous response. Motivated by the reviewer's helpful reminder, in the second revision, we further supplemented the glancing incidence XRD (GIXRD) pattern of the α -MoO₃/ η -Mo₄O₁₁ sample to provide a more comprehensive and accurate XRD analysis. Given these efforts in two rounds of revision, we believe the XRD concerns have been addressed thoroughly.

The powder XRD pattern provides information on both thin film and substrate. However, glancing incidence XRD (GIXRD) measurements can eliminate diffraction signals from the substrate by precisely controlling the penetration depth of X-rays, offering integral nondestructive information on the thin film alone.¹ The Supplementary Figure 2 displays the GIXRD pattern collected at the near surface of the α -MoO₃/ η -Mo₄O₁₁ thin film sample. As compared to its XRD pattern in Supplementary Figure 1, the GIXRD pattern exhibits additional peaks located at 13.2 and 33.8 degrees, which can be ascribed to the (001) and (-810) of η -Mo₄O₁₁ (ICSD: #15814). Moreover, the signal at approximately 12.9 degree is noticeably stronger in the GIXRD than that in the XRD, matching well with the (020)/(110) intensity ratio of the α -MoO₃ (ICSD #36167). Agreeing well with the XRD result, the GIXRD result further confirms the coexistence of α -MoO₃ and η -Mo₄O₁₁ phases in the film.

Considering all the consistent results obtained from XRD, GIXRD, Raman, XPS, HRTEM, and HAADF-STEM, we can confirm the presence of the well-constructed coherent interface between α -MoO₃ and η -Mo₄O₁₁ in the α -MoO₃/ η -Mo₄O₁₁ sample. Based on the reviewer's comment, we added the GIXRD pattern in the revised manuscript.

Supplementary Figure 1 b, The XRD patterns of α -MoO₃/ η -Mo₄O₁₁ thin films with intensity data.

Supplementary Figure 2, The GIXRD patterns of α -MoO₃/ η -Mo₄O₁₁ thin films with intensity data. (This figure is added in the Supplementary Figure 2)

Supplementary Figure 4 Raman spectra of the η - Mo_4O_{11} , α - MoO_3/η - Mo_4O_{11} , and α - MoO_3 thin films.

Supplementary Figure 8 a–c, Core-level Mo 3d XPS spectra of α - MoO_3 , α - MoO_3/η - Mo_4O_{11} , and η - Mo_4O_{11} thin films, respectively.

Fig. 3d, HAADF-STEM image for the α - MoO_3/η - Mo_4O_{11} heterostructure. The blue square corresponds to the monoclinic Mo_4O_{11} viewed along the [100] zone axis while

the red square corresponds to the orthorhombic MoO_3 viewed along the $[101]$ zone axis. **e**, The O K -edge ELNES spectra of the $\alpha\text{-MoO}_3/\eta\text{-Mo}_4\text{O}_{11}$ heterostructure obtained along the yellow arrow in the left STEM image.

Supplementary Figure 7 a, HRTEM image for the $\alpha\text{-MoO}_3/\eta\text{-Mo}_4\text{O}_{11}$ sample. **b**, HRTEM image for interface region of the $\alpha\text{-MoO}_3/\eta\text{-Mo}_4\text{O}_{11}$ sample along the $[100]$ zone axis of $\alpha\text{-MoO}_3$. FFT images for monoclinic $\eta\text{-Mo}_4\text{O}_{11}$ (collected from A area) of the $\alpha\text{-MoO}_3/\eta\text{-Mo}_4\text{O}_{11}$ sample along the $[001]$ zone axis. FFT images for layered $\alpha\text{-MoO}_3$ (collected from B area) of the $\alpha\text{-MoO}_3/\eta\text{-Mo}_4\text{O}_{11}$ sample along the $[100]$ zone axis.

Reviewer comments, third round -

Reviewer #1 (Remarks to the Author):

The manuscript tries to demonstrate the redesigned α -MoO₃/ η -Mo₄O₁₁ heterostructure which has an opportunity to avoid high lattice expansion and during electrochemical cycling. Authors claim such heterojunction enhance Li⁺ mobility and storage with longer life cycle and/or specific capacity. This manuscript is revised already two times and the text and art work has improved significantly. However, there is still a critical issue which needs clarification before this manuscript can be published.

In the revised version of the manuscript (R2), the supplementary Figure 15 shows α -MoO₃ has the almost the initial cell capacity (\sim 200mAh/g) after 1000 cycles [Figure 15 (a)]. However, the capacity of the α -MoO₃/-Mo₄O₁₁ cathode has dropped to \sim 50mAh/g already after 300 cycle. The same argument is also valid for dQ/dV profile. When the capacity loss is realized by 75% after 300 cycles, why should any industry consider this complex heterointerface for commercialization? Again, when α -MoO₃ is performing much better why is such an interface necessary at all?

Reviewer #3 (Remarks to the Author):

The authors have made several updates and clarifications to the submitted manuscript, however I am still not sure of the role and extent of the Mo₄O₁₁ as noted in the manuscript.

Checking your XRD figures you have broad peaks - which brings in other issues associated with a Rietveld refinement and the extent to which conclusions can be drawn. As an example, you refine out the %Mo₄O₁₁ as a secondary phase but the number of unique peaks (not even commenting on the low symmetry of the parent phase) is very low. You may not truly have enough data to get a good number.

MoO₃ is a layered compound - was preferred orientation an issue? was it refined?

Point-by-point response to reviewers' comments

Reviewer #1 (Remarks to the Author):

The manuscript tries to demonstrate the redesigned α -MoO₃/ η -Mo₄O₁₁ heterostructure which has an opportunity to avoid high lattice expansion and during electrochemical cycling. Authors claim such heterojunction enhance Li⁺ mobility and storage with longer life cycle and/or specific capacity. This manuscript is revised already two times and the text and art work has improved significantly. However, there is still a critical issue which needs clarification before this manuscript can be published.

Response: We sincerely appreciate the Reviewer's positive evaluation on the quality of our work after two rounds of revision. Herein, we have addressed issues and revised the manuscript following your comments.

In the revised version of the manuscript (R2), the supplementary Figure 15 shows α -MoO₃ has the almost the initial cell capacity (~200 mAh/g) after 1000 cycles [Figure 15 (a)]. However, the capacity of the α -MoO₃/ η -Mo₄O₁₁ cathode has dropped to ~50 mAh/g already after 300 cycle. The same argument is also valid for dQ/dV profile. When the capacity loss is realized by 75% after 300 cycles, why should any industry consider this complex heterointerface for commercialization? Again, when α -MoO₃ is performing much better why is such an interface necessary at all?

Response: Thanks for pointing out this critical issue. We are very sorry for the mistake in the Caption of Supplementary Figure 15 that led to your misunderstanding.

We have corrected the Caption of Supplementary Figure 15 as:

“Supplementary Figure 15 Galvanostatic charge/discharge profiles of the (a) α -MoO₃/ η -Mo₄O₁₁ and (c) α -MoO₃ cathodes collected at different cycle numbers. Calculated dQ/dV profiles of the (b) α -MoO₃/ η -Mo₄O₁₁ and (d) α -MoO₃ electrodes

after different cycle numbers at a specific current of 0.5 A g^{-1} .”

As depicted in Supplementary Figure 15a–b, the redox peaks of $\alpha\text{-MoO}_3/\eta\text{-Mo}_4\text{O}_{11}$ electrode are well retained after 1000 cycles and the corresponding dQ/dV plots at the 1st and 1000th cycles are nearly overlapped with minimal capacity loss, further demonstrating the high stability of the heterostructure with coherent interface. In sharp contrast, the $\alpha\text{-MoO}_3$ displays an irreversible capacity loss of $\sim 73\%$ after 300 cycles at a specific current of 0.5 A g^{-1} . Moreover, the redox peaks in dQ/dV plot of $\alpha\text{-MoO}_3$ are remarkably reduced and new peaks emerge even after 300 cycles, suggesting severe structural change.

We have revised the caption and carefully checked the whole manuscript to avoid similar confusion.

In the Supplementary Information (Supplementary Figure 15 in Page 16):

Supplementary Figure 15 Galvanostatic charge/discharge profiles of the (a) $\alpha\text{-MoO}_3/\eta\text{-Mo}_4\text{O}_{11}$ and (c) $\alpha\text{-MoO}_3$ cathodes collected at different cycle numbers.

Calculated dQ/dV profiles of the (b) $\alpha\text{-MoO}_3/\eta\text{-Mo}_4\text{O}_{11}$ and (d) $\alpha\text{-MoO}_3$ electrodes

after different cycle numbers at a specific current of 0.5 A g^{-1} .

Reviewer #3 (Remarks to the Author):

The authors have made several updates and clarifications to the submitted manuscript, however I am still not sure of the role and extent of the Mo_4O_{11} as noted in the manuscript.

Response: Thanks to the reviewer for helping us improve our manuscript. Concerning the role of Mo_4O_{11} , we think it has been clearly demonstrated in the present manuscript.

As presented in Fig. 4e, the cycle performances of the $\alpha\text{-MoO}_3$, $\eta\text{-Mo}_4\text{O}_{11}$, and $\alpha\text{-MoO}_3/\eta\text{-Mo}_4\text{O}_{11}$ cathodes between 1.0 and 3.5 V (vs. Li/Li^+) at a specific current of 0.5 A g^{-1} are compared. It is clear that the $\alpha\text{-MoO}_3$ cathode exhibits fast capacity fading with only approximately 27% capacity retained after 300 cycles, suggesting continuous structural degradation during cycling. Remarkably, the $\alpha\text{-MoO}_3/\eta\text{-Mo}_4\text{O}_{11}$ cathode displays both the highest specific capacity and the best cycle performance with a negligible capacity loss after 300 cycles. The superior structural stability of the heterostructure can be further demonstrated by a long-term cycling test. When cycled at a low specific current of 0.5 A g^{-1} , the $\alpha\text{-MoO}_3/\eta\text{-Mo}_4\text{O}_{11}$ cathode can retain over 90% of the initial capacity after 1000 cycles. Therefore, it is clear Mo_4O_{11} in the heterostructure can greatly improve the structural stability of $\alpha\text{-MoO}_3$, enabling notably extended cycle life for the heterostructured cathode.

Fig. 4e Cycle performance comparison of the α -MoO₃, α -MoO₃/ η -Mo₄O₁₁, and η -Mo₄O₁₁ cathodes at 0.5 A g⁻¹.

Further shown in Fig. 5a and 5b, the lattice pinning effect of Mo₄O₁₁ in the heterostructure is well demonstrated by ex situ XRD. For α -MoO₃, the gradual disappearance of the (020) peak belonging to the pristine α -MoO₃ along with the emerging (030) peak of the lithiated α -MoO₃ (Li_xMoO₃, x~0.25), at ~2.8 V vs. Li/Li⁺, confirms the irreversible phase transformation during the initial discharge process. Simultaneously, the intensive (110) and (040) peaks ascribed to α -MoO₃ vanish and a new broad peak appears in the following lithiation process, indicating the original stacking order of Mo-O octahedron layers becomes strongly disordered. In contrast to α -MoO₃, all of the characteristic peaks for the α -MoO₃/ η -Mo₄O₁₁ heterostructure are well retained with only reversible peak shift during charge/discharge, demonstrating α -MoO₃ layered structure can be stabilized without phase transition in the heterostructure. Meanwhile, the corresponding interlayered spacing of (020) planes of α -MoO₃ in heterostructure increases from 6.80 Å at 3.5 V to 6.96 Å at 1.0 V (expansion rate 2%), whereas the initial interlayer spacing of the pure α -MoO₃ exhibits a considerably increased expansion rate of 16% (from 6.80 to 7.90 Å), demonstrating the coherent interface can effectively pin lattice expansion of α -MoO₃ in heterostructure. The suppressed phase transition and lattice expansion of α -MoO₃ in heterostructure well explain its greatly improved cycle performance as compared to individual α -MoO₃.

Fig. 5 a,b Ex-situ XRD measurements of the α -MoO₃ and α -MoO₃/ η -Mo₄O₁₁ electrodes during the first discharge–charge processes in a voltage range between 1.0 and 3.5 V (vs. Li/Li⁺).

Based on the electrochemical behavior and structural evolution, the lattice pinning strategy via using Mo₄O₁₁ to stabilize α -MoO₃ in the intergrown α -MoO₃/ η -Mo₄O₁₁ heterostructure is well demonstrated in this work.

Checking your XRD figures you have broad peaks - which brings in other issues associated with a Rietveld refinement and the extent to which conclusions can be drawn. As an example, you refine out the %Mo₄O₁₁ as a secondary phase but the number of unique peaks (not even commenting on the low symmetry of the parent phase) is very low. You may not truly have enough data to get a good number.

Response: Thanks for the insightful comments. We agree with the reviewer that the relatively broad peaks of thin film XRD data may influence quantitative phase analysis by Rietveld refinement. Therefore, there could be an error for the estimated mass ratio of Mo₄O₁₁ in the heterostructure from Rietveld refinement. Nevertheless, based on the low reliability factor R_{wp} (8.3%), the error should be small and the value of mass ratio of Mo₄O₁₁ should be valid for estimation.

Despite this, the main purpose of XRD characterization is to confirm the existence of two phases (Mo₄O₁₁ and MoO₃) in the heterostructure. The two phases can be well indexed in the current XRD data, which also agrees well with other characterizations including TEM, STEM, Raman, and XPS, consolidating the

conclusion of heterostructure formation in the thin film electrode.

MoO₃ is a layered compound - was preferred orientation an issue? was it refined?

Response: Thanks to the reviewer for this insightful question. α -MoO₃ is a layered compound, and the preferred orientation of the film could affect Li⁺ ion transport kinetics (or rate performance). Based on the crystal structure of α -MoO₃ (Fig. R1), the preferred (020) orientation of the film could block Li⁺ diffusion because no open channels are provided on this plane. If the film has the preferred (020) orientation, Li⁺ ions can only diffuse into the film through grain boundaries, which definitely compromises its rate performance. Based on the XRD result (Fig. 2d), our thin film samples do not have this (020) preferred orientation and this won't be the issue for our thin film samples.

In the present work, we focus on the heterostructure design to improve the structural stability of α -MoO₃ and the influence of thin film orientation on its electrochemical performance is beyond the scope of the current study. We may carry out further research on this interesting topic in the future.

Fig. R1 Crystal structure of α -MoO₃ along the z-direction.

Reviewer comments, fourth round -

Reviewer #3 (Remarks to the Author):

thank you for the clarifications.

1) MoO₃ or its Mo₄O₁₁/MoO₃ are not really considered high energy (HE)(intro) - 180 mAh/g near 2V does not compare to the HE NMC cathodes (200 mAh/g at 3.5-4V)

2) by Mo migration (DFT) do you mean the Mo actually moves outside its octahedra or just shifts from an internal displacement to a regular internal position coordination?

3) figure 2b XRD seems like it should be under 2c XRD and vice versa.

4) The capacity is around 1Li cation (50% of theoretical) - can you get to a greater number more typical for both 4-11 and MoO₃? (250 or so?)

5) Mo₄O₁₁ has been identified it self as a good cathode (more of a channel structure by Seshadri) - with the unknown error in the amount of 4-11, could it be in higher %s than you model ?

Point-by-point response to reviewers' comments

Reviewer #3 (Remarks to the Author):

thank you for the clarifications.

1) MoO₃ or its Mo₄O₁₁/MoO₃ are not really considered high energy (HE) (intro)-180 mAh/g near 2 V does not compare to the HE NMC cathodes (200 mAh/g at 3.5–4 V)

Response: Thanks for the comments. As shown in Fig. 4d, the α -MoO₃/ η -Mo₄O₁₁ cathode exhibits a high reversible capacity of approximately **350 mAh g⁻¹ at 0.1 A g⁻¹** (~0.25 C) due to the Mo⁶⁺/Mo⁴⁺ redox couple with ~2 lithium storage per Mo. Therefore, the energy density of α -MoO₃/ η -Mo₄O₁₁ cathode materials is approximately 760 Wh/kg, which is similar to that of HE NMC (~750 Wh/kg). [*Adv. Mater.*, 2019, 31, 1900985; Zhang et al. *Nature*, 2022, 610, 67–73] More importantly, in the present work, we focus on the heterostructure design to improve the structural stability of α -MoO₃. The lattice pinning strategy could be applied to other high-energy layered oxide cathodes to achieve high structural stability and extended cycle life. We will carry out further research on this interesting topic in the future.

Fig. 4d Rate capabilities of the α -MoO₃, α -MoO₃/ η -Mo₄O₁₁, and η -Mo₄O₁₁ cathodes.

2) by Mo migration (DFT) do you mean the Mo actually moves outside its octahedra or just shifts from an internal displacement to a regular internal position coordination?

Response: Thanks for the question. The off-center displacement of d^0 Mo in MoO_6 octahedron of $\alpha\text{-MoO}_3$ (Mo^{6+}) can be ascribed to the second-order Jahn–Teller (SOJT) effect. During the lithiation process, the distorted Mo^{6+} ion in MoO_6 octahedron of $\alpha\text{-MoO}_3$ shifts from an internal displacement to the internal center of octahedron after lithiation (Fig. 6a–c). So Mo does not move outside its octahedron.

Fig. 6 Insights from theoretical calculations. a,b Differential charge density of MoO_3 and $\text{f-Li}_{0.25}\text{MoO}_3$. The blue and red regions represent the depletion and accumulation of charge, respectively. c Layered MoO_3 with Mo off-centering and the displacement of Mo ion being shifted toward the orbital center during lithiation.

3) figure 2b XRD seems like it should be under 2c XRD and vice versa.

Response: Thanks for this suggestion. However, Fig. 2b is not XRD, and Fig. 2c and 2d are XRD in original Fig. 2. If we do not understand wrongly, the reviewer suggests moving Fig. 2c under Fig. 2d. Based on this suggestion, we rearranged the XRD spectra in Fig. 2 for better comparison.

Fig. 2| Fabrication of the α -MoO₃/ η -Mo₄O₁₁ heterostructure. **a** Schematic diagram illustrating the fabrication of η -Mo₄O₁₁, α -MoO₃/ η -Mo₄O₁₁, and α -MoO₃ thin films by magnetron sputtering. **b,c** Rietveld-refined XRD patterns of the α -MoO₃/ η -Mo₄O₁₁ and α -MoO₃ thin films. The asterisk represents the substrate. **d** The corresponding O₂/Ar flow ratio versus the deposition temperature phase diagram. **e–g** Cross-section FESEM images of η -Mo₄O₁₁, α -MoO₃/ η -Mo₄O₁₁, and α -MoO₃ thin films, respectively.

4) The capacity is around 1Li cation (50% of theoretical) - can you get to a greater number more typical for both 4-11 and MoO₃? (250 or so?)

Response: Thanks for the question. The theoretical specific capacity of α -MoO₃ is about 372 mAh g⁻¹, corresponding to the Mo⁶⁺/Mo⁴⁺ redox couple with 2 lithium storage per Mo. In this work, as shown in Fig. 4d, the α -MoO₃/ η -Mo₄O₁₁ cathode can deliver a high reversible capacity of approximately 350 mAh g⁻¹ at 0.1 A g⁻¹, which is

close to the theoretical capacity for 2 Li storage.

Fig. 4d Rate capabilities of the α -MoO₃, α -MoO₃/ η -Mo₄O₁₁, and η -Mo₄O₁₁ cathodes.

5) Mo₄O₁₁ has been identified itself as a good cathode (more of a channel structure by Seshadri) - with the unknown error in the amount of 4-11, could it be in higher %s than you model?

Response: Thanks for the question. We agree with the reviewer that η -Mo₄O₁₁ could be a good cathode candidate due to its good structural stability. However, its rate performance is very poor as compared in Fig. 4d. When it is combined with α -MoO₃, the α -MoO₃/ η -Mo₄O₁₁ heterostructure exhibits both large specific capacity and good rate performance.

Deng et al. demonstrated that Rietveld quantitative phase analysis (QPA) can yield accurate results, with errors generally less than 2.0% absolute. [J Forensic Sci, 2015, 60, 1040–1045] In the present work, the refined phase fraction error for η -Mo₄O₁₁ in the heterostructure is 1.3% as provided by the MAUD. Therefore, the real phase ratio of Mo₄O₁₁ in the heterostructure could either be slightly higher or lower than that for the refinement model.

Fig. 4d Rate capabilities of the $\alpha\text{-MoO}_3$, $\alpha\text{-MoO}_3/\eta\text{-Mo}_4\text{O}_{11}$, and $\eta\text{-Mo}_4\text{O}_{11}$ cathodes.